# Reconciling contrasting views on economic complexity

Carla Sciarra 🔟 [1✉], Guido Chiarotti[1], Luca Ridolfi[1] & Francesco Laio[1]

Summarising the complexity of a country's economy in a single number is the holy grail for scholars engaging in data-based economics. In a field where the Gross Domestic Product remains the preferred indicator for many, economic complexity measures, aiming at uncovering the productive knowledge of countries, have been stirring the pot in the past few years. The commonly used methodologies to measure economic complexity produce contrasting results, undermining their acceptance and applications. Here we show that these methodologies – apparently conflicting on fundamental aspects – can be reconciled by adopting a neat mathematical perspective based on linear-algebra tools within a bipartite-networks framework. The obtained results shed new light on the potential of economic complexity to trace and forecast countries' innovation potential and to interpret the temporal dynamics of economic growth, possibly paving the way to a micro-foundation of the field.

[1] DIATI, Politecnico di Torino, Corso Duca degli Abruzzi 24, 10129 Torino, Italy. ✉email: carla.sciarra@polito.it

The French politician and gastronome Jean Anthelme Brillat-Savarin in his book 'Physiologie du goût' wrote: "Tell me what you eat, and I will tell you who you are"[1], aphorising on the fact that people's food baskets reflect their wealth status. In the same vein, metrics of economic complexity (EC) aim at defining the socio-economic status of countries grounded on their export baskets[2]. Within economics, these approaches mainly serve as an alternative to more traditional economic growth theories[3–8] which are often blamed for shrinking the intricacy of countries' socio-economic dynamics through simplistic assumptions[9,10].

Within this new class of metrics, the productive knowledge owned by each country – which embeds capabilities, finance, technology, human capital and resources, and determines the country's potential for economic growth – can be extracted from easy-to-find data[11]. Not surprisingly, a first proxy of countries' productive knowledge is the number of products in their export baskets, i.e., their production diversity[11–13]. Although insufficient, since it does not account for baskets' composition and complexity, the diversity is a necessary and relevant information to understand the trading competitiveness of countries[11,14]. The methodologies of economic complexity aim at improving this most obvious measure of competitiveness exploiting the information related to the sophistication of the exports and the capabilities required to produce and export a given good: countries with low productive knowledge only produce and export fewer and less sophisticated products, resulting in lower stages of competition[11,15], while more competitive countries exploit their know-how and resources to diversify their export baskets[11,15]. By reversing this reasoning, it is thus expected that the diversification and composition of the export basket can be used to measure the countries' and products' economic complexity, thus posing the bases for a data-based (bottom-up) ranking of countries and products. This rationale lies at the base of the commonly used methodologies to measure the economic complexity of countries and products, namely the Method of Reflections (MR)[12] and the Fitness and Complexity algorithm (FC)[15]. In spite of their common root, these two methods radically differ in the conceptual approach to the problem and, as a consequence, in the obtained outcomes.

The MR approach measures a country's economic complexity as the average of the complexities of the products in its export basket. In a specular manner – from which the name "reflections" –, a product's complexity is obtained as the average of the complexities of the countries exporting it. The equations defining the two averages are coupled to obtain the Economic Complexity Index (ECI) and the Product Complexity Index (PCI), which have been shown to be the result of a linear algebra exercise[11,16,17]. As an effect of taking the averages, the obtained measures turn out to lose information about countries' diversification and products' ubiquity[18] (ubiquity is defined as the number of exporters for a given product[11]) .

In contrast, Tacchella et al.[15] counter on the assumption of a linear relation between the products' and countries' complexities. In their view, the fact that a less competitive country exports a given product should unavoidably downgrade the product's complexity, an effect that the Authors argue could only be obtained through the use of a non-linear relation. As a consequence, these authors introduce two metrics, the Fitness of countries $F_c$ and the Quality of products $Q_p$, where products' Quality non-linearly depends on the Fitness of the exporting countries (see Methods section, Eq. (11)); in contrast, the Fitness is obtained as the sum of the Qualities of the exported products. In this approach, contrarily to MR, the countries' Fitness preserves the information on the diversification of the export baskets[14,19].

It is not only the mathematics of the two approaches which is different, but also the obtained outcomes significantly diverge: as shown in Supplementary Fig. 1, the countries' rankings obtained with $ECI_c$ and $F_c$ widely scatter (see Supplementary Note 1 for details on the implementation of the two algorithms). This poses an issue of practical use of the economic complexity measures, potentially undermining the very essence of the economic complexity theory. We argue that the role played by EC measures in economics and policy making (see, e.g., refs. [20–24]) requires more consistency in the outcomes of different methods.

In this paper, we reconcile the MR and FC approaches by recasting them into a mathematically-sound, multidimensional framework, which allows us to recover and combine the strengths of both methods, still maintaining the relevant feature of providing countries' and products' rankings.

## Results

**A general framework for economic complexity.** Economic complexity approaches are grounded on the trade data collected into a bipartite network, defining exporters and products, and detailing whether and how much (in monetary value) a country exports a given product. The bipartite network is interpreted as the compact representation of the tripartite network constituted by countries-capabilities-products[12,15]; most applications[12,15] take into account only the relevant exporters in the network, where the relevance is computed according to the Relative Comparative Advantage (RCA)[25]. Moreover, to highlight the role played by network's topology, the weights in the bipartite network are typically neglected, turning to a binary incidence matrix **M** where $M_{cp} = 1$ implies that the country $c$ is a relevant exporter of the product $p$ (see Methods section, Eq. (6)).

In a general framework, economic complexity theories aim at determining two properties $X_c$ and $Y_p$ – describing the complexity of country $c$ and product $p$, respectively – by a system of coupled equations

$$\begin{cases} X_c = f(Y_1, Y_2, ..., Y_p, M_{cp}), & p = [1, ..., P], \\ Y_p = g(X_1, X_2, ..., X_c, M_{cp}), & c = [1, ..., C], \end{cases} \quad (1)$$

where $f$ and $g$ are linear functions and $C$ and $P$ are the number of countries and products considered in the analysis, respectively. To consider $f$ and $g$ as linear functions allows one to recast the determination of $X_c$ and $Y_p$ as the solutions of an eigen-problem of a suitable (approach dependent) transformation matrix **W**, whose elements $W_{cp}$ are derived from **M**. In this case, these properties' values are obtained from the following coupled linear equations:

$$\begin{cases} X_c = \frac{1}{\sqrt{\lambda}} \sum_p W_{cp} Y_p, \\ Y_p = \frac{1}{\sqrt{\lambda}} \sum_c W_{cp} X_c, \end{cases} \quad (2)$$

being $\lambda$ the eigenvalue of the equivalent eigen-problem, such that the following relations hold

$$X_c = \frac{1}{\lambda} \sum_c \sum_{c^*} W_{cp} W_{c^*p} X_{c^*} = \frac{1}{\lambda} \sum_{c^*} N_{cc^*} X_{c^*}, \quad (3)$$

and

$$Y_p = \frac{1}{\lambda} \sum_p \sum_{p^*} W_{cp^*} W_{cp} Y_{p^*} = \frac{1}{\lambda} \sum_{p^*} G_{pp^*} Y_{p^*}. \quad (4)$$

A by-product of Eqs. (3)–(4) is that the squared, symmetric matrices $\mathbf{N} = \mathbf{WW^T}$ and $\mathbf{G} = \mathbf{W^TW}$ can be interpreted as proximity matrices for nations and products, respectively, where proximity defines similarity (for example, $N_{cc^*} = N_{c^*c}$ describes the similarity in the export baskets between countries $c$ and $c^*$, see Discussion section). Note that the set of equations in Eq. (2) involves the same transformation matrix **W**. This entails that: the

matrix $\mathbf{W}$ represents a weighted incidence matrix of an undirected bipartite network uniquely describing the relations between countries and products – this would no longer be true if two different matrices were used for the transformation; moreover, the feature of symmetry for the matrices $\mathbf{N}$ and $\mathbf{G}$ is essential to interpret them as proximity matrices, thus defining a bijective function.

The eigen-problems in Eqs. (3)–(4) have multiple solutions, provided by the eigenvalues $\lambda_i$ and the corresponding eigenvectors of the matrices $\mathbf{N}$ and $\mathbf{G}$, respectively[26]. In most situations, the eigenvector corresponding to the largest eigenvalue $\lambda_1$ carries the maximum amount of information[27] and it is thus taken as solution (although we will demonstrate the potential of combining more eigenvectors). In complex network jargon, $X_c$ and $Y_p$ are (eigen-)centrality metrics in the bipartite network of countries and products[19,28].

We now provide two examples of application of this general framework pertaining with the two aforementioned EC metrics, MR and FC, referring to these examples by using the superscripts $A$ and $B$, respectively. The MR method is simply recast by setting $W_{cp}^A = M_{cp}/\sqrt{k_c k_p}$ in Eq. (2), which provides the indices $ECI_c$ and $PCI_p$ from the transformations $X_{c,2}^A = ECI_c \sqrt{k_c}$ and $Y_{p,2}^A = PCI_p \sqrt{k_p}$ – being $k_c = \Sigma_p M_{cp}$ the degree of the countries, i.e., their diversity, and $k_p = \Sigma_c M_{cp}$ the degree of the products, i.e., their ubiquity. In this case, the first eigenvectors $X_{c,1}^A$ and $Y_{p,1}^A$ carry a trivial information, since they equal the square roots of the degrees, $k_c$ and $k_p$ (see Methods section, Eq. (8)), thus leading to unitary $ECI_c$ and $PCI_p$ values, discarded in the original works for being uninformative[11,12]. For this reason the eigenvectors $X_{c,2}^A$ and $Y_{p,2}^A$, corresponding to the second largest eigenvalue, are taken by the authors as the solution of Eqs. (3)–(4)[18]. The mapping $\{ECI_c, PCI_p\} \Leftrightarrow \{X_{c,2}, Y_{p,2}\}$ completely preserves the MR outcoming information.

Instead, the FC method is recast by setting $W_{cp}^B = M_{cp}/k_c k_p'$, $X_{c,1}^B = F_c/k_c$, and $Y_{p,1}^B = Q_p k_p'$ in Eq. (2), where $k_p' = \sum_c M_{cp}/k_c$ (see Methods section, Eq. (13)). Differently from the MR mapping, in the case of FC, this mapping is not merely the results of algebraic manipulation, but implies a non-trivial linearisation of the relation between Quality and Fitness values (see Methods section, Eqs. (11)–(13)). Surprisingly enough, comparing the terms $X_{c,1}^B$ and $F_c/k_c$, or $X_{c,1}^B k_c$ and $F_c$, for the Fitness values – analogously $Y_{p,1}^B$ and $Q_p k_p'$ (or $Y_{p,1}^B/k_p'$ and $Q_p$) for the Quality values – this linearisation almost entirely preserves the information of the non-linearly computed values (independently of the kind of indicator used to measure correlation, Supplementary Fig. 2). Notice also that our linear formulation does not suffer from the well-known convergence problems of the iterative FC algorithm[29] and provides more regular solutions.

Some comments on the obtained results are due to the reader. First, the original $ECI_c$, $PCI_c$, $F_c$ and $Q_p$ variables are recovered within our general framework through simple (but non-trivial) mappings from $X_c$ and $Y_p$. The use of the variables $X_c$ and $Y_p$ allows one to gain neatness in the mathematics, reflected by the fact that the matrices $\mathbf{N}$ and $\mathbf{G}$ can be considered as suitable proximity matrices containing information about the similarities among countries and products, respectively. This aspect may have important consequences on the interpretation of the economic significance of these metrics, as outlined in the Discussion section. Second, the matrices $\mathbf{W}^A$ and $\mathbf{W}^B$ differ for the specific scaling factors adopted on the matrix $\mathbf{M}$. It is hard to recognise an economic (or a mathematical) basis on how the factors are determined, and this leaves no solid ground for a potential user to

decide which approach, between MR and FC, to follow. Third, notwithstanding the differences among $\mathbf{W}^A$ and $\mathbf{W}^B$, the eigenvectors $X_{c,1}^A$ and $X_{c,1}^B$ carry similar information (Supplementary Fig. 3), as also $X_{c,2}^A$ and $X_{c,2}^B$ (this is also partially true for $Y_p$, Supplementary Fig. 4). Therefore, the divergences between $F_c$ and $ECI_c$ – and corresponding outcomes – shown in Supplementary Fig. 1 should be mainly attributed to the fact that eigenvectors of different order are considered in the two approaches. Hence, the two metrics bring different information; albeit different, this information is relevant for both metrics, as demonstrated by numerous practical applications of the two approaches[20–22,24,30–33].

Grounded on these considerations, we promote here an integrated measure of economic complexity, which exploits the neatness of the proposed framework. By employing the recently introduced framework to deal with multidimensional centrality[28], we combine the two existing measures into unique centrality metrics unveiling the multidimensional complexity of countries and products. Either using $\mathbf{W}^A$ or $\mathbf{W}^B$ to develop the new integrated measure of complexity would lead to reliable and comparable results. We lean toward the use of $\mathbf{W}^B$, the one related to the FC method, for the following reasons: on the one hand, the first eigenvector $X_{c,1}^A$ – from which, using Eq. (8), the unitary first eigenvector of MR is recovered – equals $\sqrt{k_c}$, thus carrying no added information beyond diversity (and the same holds for products); on the other hand, the last update on the MR method, named ECI+[34], has been shown to be equivalent to the non-linear FC algorithm[35], thus implicitly supporting the idea that FC carries more information then MR. The grounding hypotheses about the hidden capabilities of countries and on how these can be deducted by looking at the export baskets of countries upon which the EC algorithms are built - are preserved in our framework. From here on, we will thus use the matrix $\mathbf{W}^B$ in Eq. (2) and drop the superscript $B$ in the mathematical notation. In the following, we will focus on the analysis of countries' complexity. A similar reasoning also applies to the sophistication of products, whose details for the computation are given in the Methods section, Eqs. (18)–(19), while results are shown in Supplementary Figs. 4–5 and commented in Supplementary Note 2.

**The generalised economic complexity index.** We propose to distil the information on economic complexity into a GENeralised Economic comPlexitY index, GENEPY (the Genepy is a herb-based distillate typical of the north-western part of Italy). The GENEPY index for countries is defined as follows:

$$GENEPY_c = \left( \sum_{i=1}^{2} \lambda_i X_{c,i}^2 \right)^2 + 2\sum_{i=1}^{2} \lambda_i^2 X_{c,i}^2, \qquad (5)$$

where $X_{c,1}$ and $X_{c,2}$ are the eigenvectors corresponding to the first two largest eigenvalues $\lambda_1$ and $\lambda_2$ of the proximity matrix

$$\begin{cases} N_{cc^*} = \sum_p W_{cp} W_{c^* p} = \sum_p \dfrac{M_{cp} M_{c^* p}}{k_c k_{c^*} (k_p')^2}, & \text{if } c \neq c^*, \\ N_{cc^*} = 0, & \text{if } c = c^*, \end{cases}$$

in which the redundant information of the self-proximity is deleted setting all diagonal elements to an arbitrary constant value (we set this value to zero). The rationale to compute the GENEPY index grounds on two key points: firstly, to interpret the symmetric squared matrix $\mathbf{N}$ as the mathematical description of the weighted topology of an undirected network[26] – such that the countries are the nodes and the similarities between the export baskets are the links connecting them – and, secondly, to interpret the eigenvectors of $\mathbf{N}$ as the (multidimensional) eigenvector centrality of the nodes in the network. Using this approach, the

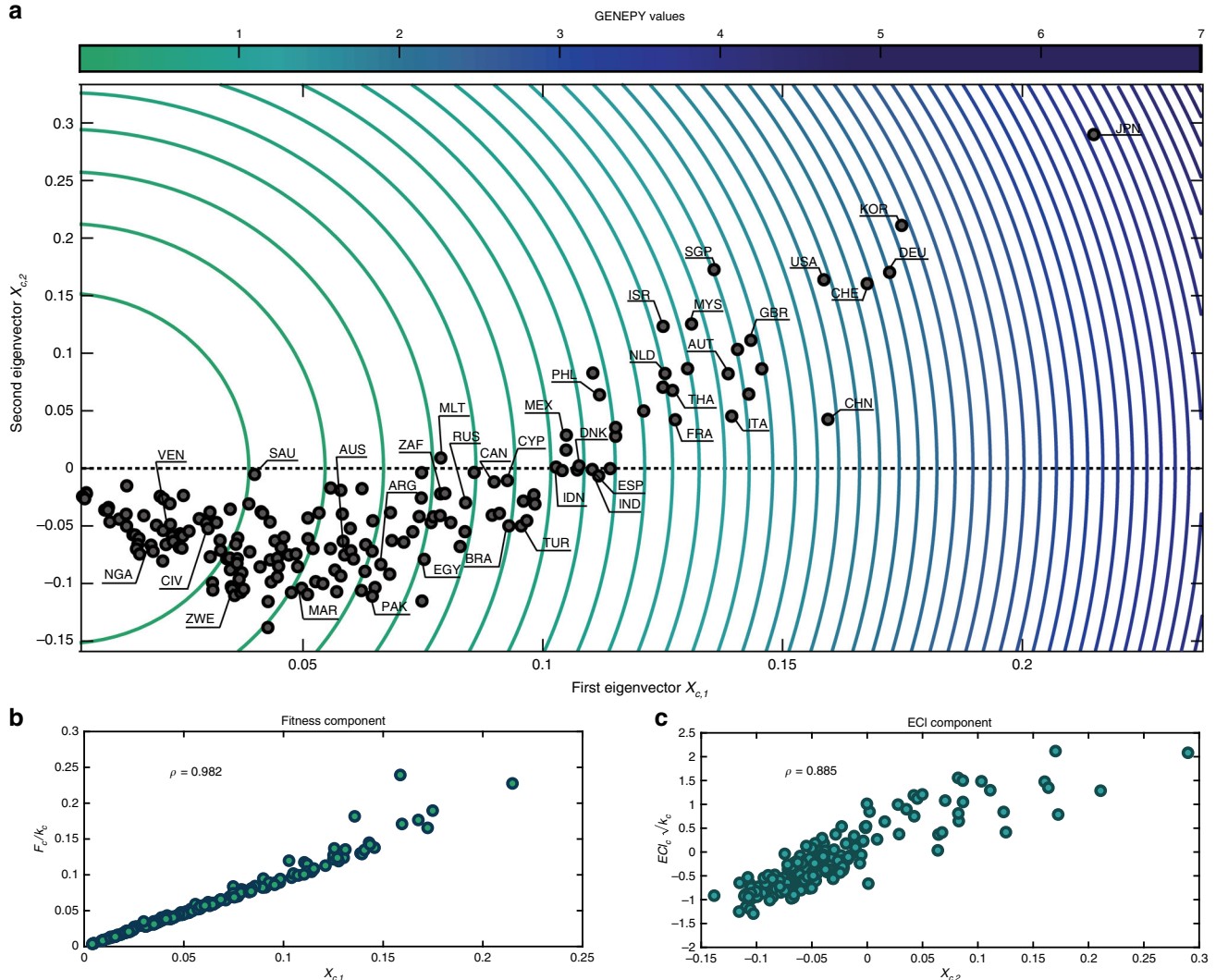

**Fig. 1 The GENEPY index and its components. a** $\{X_{c,1}, X_{c,2}\}$ plane and $GENEPY_c$ from the data of 2017 international products' trade. The x-axis reports the components of the first eigenvector $X_{c,1}$, whilst the y-axis the components of the second eigenvector $X_{c,2}$. The eigenvectors are normalised such that their Frobenius norm is unitary, i.e., $\sum_c X_{c,1}^2 = \sum_c X_{c,2}^2 = 1$. Contours range from lower $GENEPY_c$ values (green) to higher ones (blue). **b** Fitness component. Scatter plot of the first component $X_{c,1}$ compared with the values of the Fitness values $F_c$ rescaled by the countries degree $k_c$ (see Methods section, Eq. (14)). **c** ECI component. Scatter plot of the second component $X_{c,2}$ compared with $ECI_c$ values rescaled by the term $\sqrt{k_c}$ (see Methods section, Eq. (8)). The correlation coefficient in the plots **b** and **c** is of the Pearson's kind. Figures have been produced with MATLAB 2019b.

eigenvectors are combined into a unique metrics (the GENEPY one), following a statistically grounded framework where the same eigenvectors are obtained as the result of a least-squares estimation exercise[28] (Methods section, Eqs. (18)–(19); for more details we refer the reader to Sciarra et al.[28]).

We exemplify the use of the GENEPY index by considering the international trade of goods during the years 1995–2017[36]. In Fig. 1, the results are processed for the 2017 trade. Figure 1a displays the position of countries on the $\{X_{c,1}, X_{c,2}\}$ plane. Most economies with a high drive for innovation and technology[37] – such as the UE-28 countries, Switzerland (CHE), China (CHN), Japan (JPN), Singapore (SGP) and the United States of America (USA) – are found far from the origin. This entails the presence of top-quality products among their exports and, therefore, of relevant productive knowledge. Less economically stable economies, such as those of many African and South-American countries, are located in the bottom-left part of the graph. The GENEPY index also identifies potentially top-competitive countries, such as Australia (AUS) and Canada (CAN), struggling to

boost their complexity due to remoteness and resources-dependency, well-known factors for affecting trade and economic growth[38–40]. The information distilled through the GENEPY index can be better understood by considering the meaning of its components, i.e., the two eigenvectors $X_{c,1}$ and $X_{c,2}$, as contextualised in complex network theory[26]. In fact, the elements of the first eigenvector represent the eigenvector centrality of the countries as obtained from the proximity matrix $\mathbf{N}$, interpreting the matrix as the weighted, adjacency matrix of an undirected network connecting the countries for the similarities in their export baskets (see Discussion section). Instead, the values of $X_{c,2}$ cluster countries according to the similarities in their export baskets. In fact, the strict nexus between $X_{c,2}$ and $ECI_c$ recalls the results provided in Mealy et al.[41], where the Authors proved that ECI perfectly solves a spectral clustering algorithm. Interpreting this result within the network of similarities designed by $\mathbf{N}$, the GENEPY centrality identifies that set of capabilities (contributing to the productive knowledge) a country owns and shares with others. In this sense, more central nodes are found within a

cluster of highly competitive countries, while less complex countries are found moving towards the borders of the graph. This result is confirmed by the reordering of the matrix **N** according to the GENEPY values (Supplementary Fig. 6, Supplementary Table 1), showing that countries with higher complexity share similar sets of capabilities, as their export baskets are similar.

As mentioned, our framework combines the advantages – and information – of the two existing metrics of economic complexity, ECI and Fitness. On the one hand, the countries' Fitness values obtained with the iterative FC method are recovered, with great accuracy, from the product of the first eigenvector $X_{c,1}$ with $k_c$ (see Fig. 1b and Methods section, Eq. (14)). The very small deviations from the 1:1 line shown in Fig. 1b are not induced by the linearisation procedure. In fact, they disappear when the equation $N_{cc^*} = \sum_p M_{cp} M_{c^*p}/k_c k_{c^*} (k_p')^2$ is used also for $c = c^*$, i.e., when the matrix **N** is not interpreted as a proximity matrix (Methods section, Eq. (15) and Supplementary Fig. 7). However, this would imply inflating the $F_c$ (or $X_{c,1}$) values for countries with large self-interactions, which, in our opinion, induces an undesired bias in the results. On the other hand, a good proxy of the $ECI_c$ values is obtained by dividing the values of the second eigenvector $X_{c,2}$ by $\sqrt{k_c}$, as shown in Fig. 1c (Methods section, Eq. (8)). In this case, the scatter of the plot is due to the differences in the matrices $\mathbf{N}^A$ and $\mathbf{N}^B$ (Methods section, Eqs. (9) and (15)), respectively.

Being the GENEPY framework grounded on both existing indicators of economic complexity (the FC and MR algorithms), it inherits the intuitions and rationales upon which these two metrics are built: the capabilities of countries to export diversely complex goods are hidden within the bipartite network of countries and exports, under which they combine to maximise the complexity of the goods. Also, since $X_{c,1}$ maintains a very high correlation with $k_c$ (Supplementary Fig. 8), our framework preserves the information on diversification, which is a relevant one to understand how export capabilities are exploited by countries.

## The trajectories of economic growth

The ability of the proposed multidimensional index to assess the sophistication of countries' export-baskets and, simultaneously, define clusters of economic growth can be exploited to track the path toward prosperity of countries as driven by economic complexity. In fact, according to the economic complexity theory, a country's acquisition of capabilities, employed in the production – and hence export – of goods[2,11,42] is a determining factor for its economic growth. Any country at a lower stage of growth uses its increasing capabilities to fill its export basket with higher-quality goods, possibly similar to those traded by countries at higher stages of growth. This entails the creation of a wider export basket allowing the country to gain momentum in the market. Also, in order to boost its economic complexity – and growth – such a country may enlarge its offer including products for which it can be considered the only relevant exporter, hence gaining advantage[4]. Connecting the $GENEPY_c$ values of countries in time allows one to draw the path along this growth process, as shown in Fig. 2, in which we show some economic complexity growth paths such as the ones of China (CHN), Germany (DEU), Japan (JPN), Nigeria (NGA) and Philippines (PHL). One recognises that also the ensemble of the trajectories is knee-shaped: in fact, in each year of analysis the positions of countries in the plane $X_{c,1}$–$X_{c,2}$ arrange in a knee-like shape as shown in Fig. 1a for the year 2017. The presence of this shape is related to linear algebra and network science (Supplementary Note 3). By analysing the aggregated displacements of countries in time from 1995 to 2017 (for details

see Supplementary Fig. 9, Supplementary Note 4), it is possible to identify in the graph three regimes of growth. The first one is the "Impasse". The countries that lie within this area averagely exhibit a horizontal displacement within the borders delimited by low values of $X_{c,1}$ and negative values of $X_{c,2}$. Countries whose dynamics of growth lie in this area may suffer from lack of skills, human and capital investments and resources, thus resulting in low productive knowledge and, consequently, reduced diversification and complexity[4]. These countries hence face an impasse condition, resulting in a saddle point of growth and poor growth potential. In Fig. 2, Nigeria (NGA) and Venezuela (VEN) are tangled in this bottom-left part of the graph. The second regime is the "Bounce". It is marked by the crossing of the zero value of the y-axis and this area defines the increment in quantity and quality of the exports. Here, the average dynamics of the countries is uplifting toward higher stages of growth. Countries such as China (CHN), India (IND) and Singapore (SGP) have clearly boosted their complexity to higher levels during the last years, joining the rich countries cluster ($X_{c,2} > 0$) during the period of observation (1995–2017). The third regime is the "Arena". Once in the economically advanced cluster, countries can play in the arena of competitiveness, where the GENEPY index of some countries increases in time, that of others follows a decreasing path, instead. In fact, in this area countries aim at increasing the sophistication and the quantity of their exports which contribute to the increase of the $X_{c,1}$ values; at the same time, countries compete to become leaders in the economically grown group, hence earning scores on $X_{c,2}$. However, the entrance of new countries in the competitive market is likely to affect other countries' growth. This area includes Japan (JPN), USA, Germany (DEU) and Switzerland (CHE) as paradigmatic examples.

Therefore, during their economic growth process, countries tend to move from lower stages of complexity, delimited within the bottom-left quadrant, to higher ones, framed into the top-right quadrant. The former stage is associated with low productive knowledge and, consequently, low diversity in the exports. Contrarily, the latter is characterised by gain in skills and capital's investments, for which competition and growth are determined.

In Fig. 2, the interactions among countries are also evident. The rapid growth of a country, such as the dynamics shown by China[37,43], naturally impacts other economies, whose $GENEPY_c$ values change according to the increased complexity of the competitor. An example is given by the nested trajectories of the arena-countries, such as Germany, Japan and USA, concurrent with the raise of China and Singapore. Some steadiness points in the trajectories can also be explained by the economic history of the countries. For example, the reduced trade capacity of countries, as a consequence of the 2008 financial crisis[44], produces a drop in complexity as shown by Germany, Italy and USA among the others. Instead, the Chinese last downgrading points of 2016–2017 may be explained as spillover effects of the 2015 stock market crash[45] and could also be related to the largely debated hard landing of the Chinese economy of the last years[46].

To collapse the information on how countries' rankings evolve in time we compute, for each year in the period 1995–2017, the world's centre of GENEPY by weighting all countries' geographical barycentres by their $GENEPY_c$ values. This computation has been executed according to the procedure defined by the McKinsey Global Institute in Dobbs et al.[47] to compute the shift during history of the Gross Domestic Product (GDP); the outcomes are shown in Fig. 3 in yellow. For comparison, in Fig. 3 we replicate the same procedure to compute the trajectories of the world's barycentre by weighting the countries' barycentres by their GDP at Purchasing Power Parity (GDP PPP in blue) – and, alternatively, their population (in purple). Since the economic complexity metrics are intensive ones (i.e., their values are "per

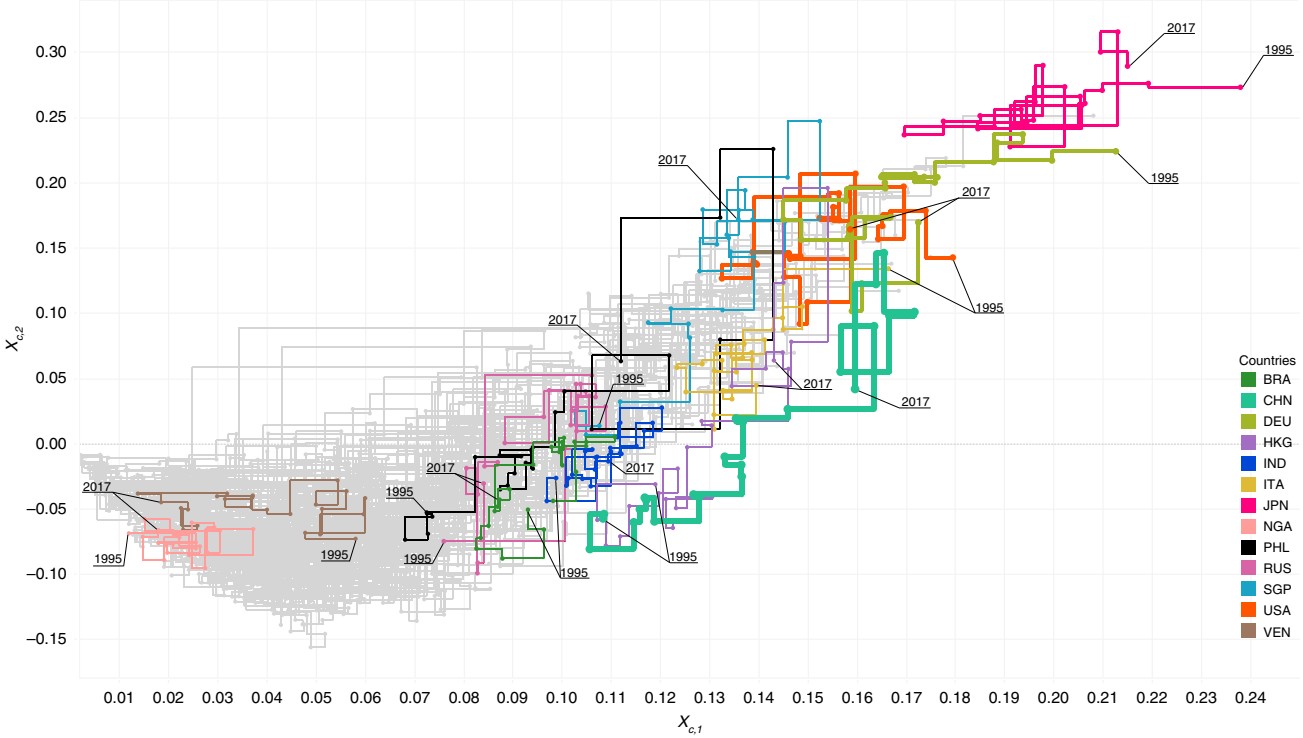

**Fig. 2 Countries' trajectories in the GENEPY plane.** The values of the first eigenvector $X_{c,1}$ are on the x-axis, whilst on the y-axis the values of the second eigenvector $X_{c,2}$ are found. The eigenvectors are normalised such that their Frobenius norm is unitary, i.e., $\sum_c X_{c,1}^2 = \sum_c X_{c,2}^2 = 1$. We highlight the trajectories of Brasil (BRA), China (CHN), Germany (DEU), Hong Kong (HKG), India (IND), Italy (ITA), Japan (JPN), Nigeria (NGA), Philippines (PHL), Russia (RUS), Singapore (SGP), United States of America (USA) and Venezuela (VEN), against the background created by trajectories of all other countries in grey. Line width reflects the countries' share of world exports in monetary value during 2017. To improve the readability of the plot, the paths from one point to another were forced to follow right-angled movements. The figure has been produced with Tableau Public 2019.4.

capita" ones[12,15,18,41]), the shifting in the world's centre of GENEPY has been computed by multiplying each country's GENEPY index for its population value in time, thus allowing for a fair comparison with the path followed by the GDP (in absolute value) in time. As the figure shows, the trajectories of the GDP and GENEPY index, differently from the population path, move towards East. The world's centre according to population, although clearly centred in the middle of Asia (as it would have been expected due to the high density of population this area has always recorded[48]), curves toward West as provoked by the increasing population in Africa[48]. The differences in the world's GENEPY, GDP and population paths confirm that, year by year, the economy is more centred in the East and that increasing population poorly impacts the ability of countries to economically grow. The distance between the current position of the barycenter of GDP and GENEPY may also imply that Asian countries (China included) still have a strong potential for economic growth, as also stated in Cristelli et al.[23]. Also, the trajectory drawn using GDP differs from the one drawn using the GENEPY index of countries as weights, because of the ability of the latter to capture both the productive knowledge of countries and the aforementioned dynamics of growth and competition between the actors in the trade.

## Discussion
We have introduced the GENeralised Economic comPlexitY index (GENEPY), which provides a multidimensional metrics of countries' (and products') complexity. GENEPY arises from the eigenvectors of a symmetric proximity matrix, describing the similarities in the export baskets of countries. These eigenvectors

combine in a multidimensional fashion, the information obtained from MR and FC metrics, thanks to a mapping (and linearisation for FC) of the original metrics to reduce the problem of finding these metrics to an eigen-centrality problem. GENEPY ranks countries for their multidimensional complexity, squeezing the eigenvectors through the adoption of a statistical framework on centrality metrics[28]. Moreover, the multidimensionality of our approach can be exploited to trace the economic growth process of countries in time. The richness of the proposed framework demands a deeper focus on some of its aspects.

A key point is that the proximity matrix **N** among countries is symmetric; as a consequence, the left and right eigenvectors coincide and the eigenvector centrality, whereupon our metrics are grounded, is distinctly defined[26,27]. In contrast, by adopting the mathematical approaches of MR or FC, asymmetric matrices are recovered to map countries' Economic Complexity (see Methods section, Eq. (7) for the MR case) – or Fitness (see Methods section, Eq. (11) for the FC case) – onto itself (a mirror argument holds for products). In this case, the eigen-problem can be formulated by considering either right or left eigenvectors, thus posing the question of how the problem should be tackled. This is not just a matter of mathematical formalism: in fact, the eigenvector centrality for directed networks – whose adjacency matrices are asymmetric – typically considers the right and left eigenvectors for determining the out and in centralities of the nodes, respectively, as caused by directionality of the edges[26]. In the same vein, the well-known PageRank[49] centrality algorithm for directed networks considers the left eigenvector to assess only the in-centrality of the nodes. For bipartite networks, the most basic and simple case to rank nodes would be to set $M_{cp} = W_{cp}$, thus providing two symmetric proximity matrices $\mathbf{M} \cdot \mathbf{M}^{\mathbf{T}}$ in Eq.

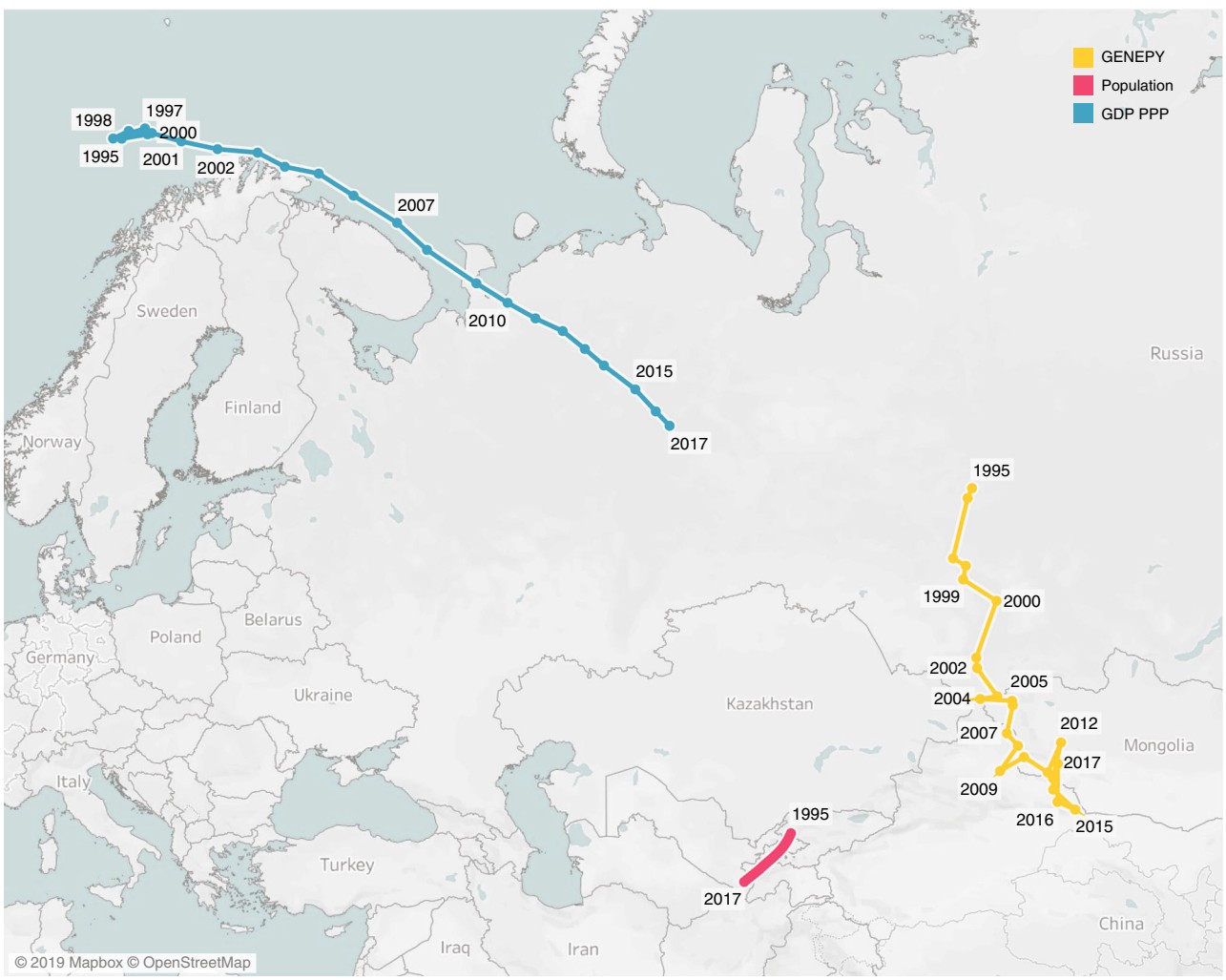

**Fig. 3 The world's economic and demographic barycentre, 1995–2017.** The trajectories are computed by weighting the countries' geographical centres by their GENEPY index, in yellow, the Gross Domestic Product at Purchasing Power Parity (GDP PPP), in blue, and the population size, in purple. The GDP trajectory is consistent with the one shown by the McKinsey Global Institute[47] taking as reference the path in there shown from 1990–2025. Data for the GDP PPP and the population of countries are provided by the World Bank. The coordinates of countries are provided by the Portland State University and defined according to the georeference system WGS 1984. The figure has been produced with Tableau Public 2019.4.

(3) and $\mathbf{M^T \cdot M}$ in Eq. (4)[50]. Contrarily, although set in a bipartite network framework, economic complexity methods as MR and FC generate artificial asymmetry by rescaling this symmetric matrices (using the countries' degree or some of its transforms) without taking care of preserving the feature of symmetry; thus leaving almost arbitrary choice to the solution of the eigen-problem. The symmetry of the transition matrices, also in terms of the adherence to the original symmetric structure of the problem, represents an added value of our framework. Moreover, the bilateral information of the proximity matrix can be used to understand the structure of the export baskets of countries and how these are related through shared common capabilities (Supplementary Fig. 6).

We have also shown how GENEPY can be used to track the economic growth of countries during the years as driven by their economic complexity. Even though economic complexity metrics have already been used to draw these paths[42,51–53], our innovative multidimensional approach allows one to draw these trajectories without the need of embedding the exogenous information on the GDP per capita that most applications require. As such, the chance of maintaining the simplicity of a data-driven approach endows the GENEPY framework with the main founding reason

for which economic complexity theory was born, i.e., to provide the ground for a more quantitative, data-driven approach to the assessment of the potential economic growth of countries as factored by the productive knowledge[54].

A further advantage of the GENEPY index is given by its robustness. In fact, when conceiving the bipartite network of countries and products, the commonly used binarisation proce-dure of the **RCA** matrix (see Methods section, Eq. (6)) is adopted, aiming at capturing the network topology. However, a different (but possibly relevant) matrix is the one obtained by directly working on the **RCA** matrix, without reverting the weighted network into a directed one. We show that, also if this path is followed, the GENEPY results remain coherent with respect to changes in the incidence matrix of the network (Supplementary Fig. 10). This does not hold when the MR and FC approaches are used.

The fact of having found very similar results between the linear and the non-linear versions of the FC algorithm (on average, 99.5% Pearson's correlation, Supplementary Fig. 2) cannot be systematically generalised to other cases: in fact, some bipartite systems may require a genuine non-linear approach to let their nested nature emerge (see, e.g., the results pertaining to the

pollinators-plants bipartite network in Supplementary Fig. 11, discussed in Supplementary Note 5). However, the good results obtained in this case suggest that there are also systems where non-linearity plays a minor role. We speculate that this might be related to the differences in the decision-making processes ruling these systems. On the one hand, e.g., nested ecological networks self-organise following ecological rules of non-linear population dynamics[55]. These systems are thus driven by more rigid decision-making processes. On the other hand, the plastic human decision-making process – which is of course at the base of the trade network self-organisation – may give rise to less nested network structures: for a given productive knowledge, trade may follow a simpler sum rule, i.e., "the more, the better", as trade enhances growth[56]; thus clarifying the reason why the diversity of a country is used as a first proxy of the productive knowledge itself.

Moreover, in the FC algorithm the Quality of a product is mainly determined by the least fit country exporting it, a crucial property accomplished by the non-linearity of the FC approach. In our linear framework, this property is maintained through the term $k'_p = \sum_c M_{cp}/k_c$, occurring in $W_{cp} = M_{cp}/k_c k'_p$. This term in fact represents the degree of a product corrected by how easily it is found within the network. Its inverse $1/k'_p$ is an anti-centrality score for the product, determining how limited is its presence within the producers' baskets and thus suggesting the need for higher productive knowledge in its production process. Notice that, by substituting the incidence matrix $\mathbf{M}$ with the traded monetary values, the term $k'_p$ also recurs in the so-called EXPY rationale by Hausmann et al.[2]. Based on a decision-making model of firms' investment choices, Hausmann et al.[2] defined an index of economic growth potential of countries, assessed through the required productive level of the exported products, i.e., EXPY. As we show, (see Methods section, Eq. (17)), the equations to compute $X_c$ in the GENEPY framework are similar to those defining the EXPY scores of countries[2]. Clearly, EXPY has been defined from a different deductive rationale, which considers the trade as described by the weighted incidence matrix of the monetary fluxes (thus providing different input information) and embeds exogenous information such as the GDP per capita. Notwithstanding these differences, the formal similarity of GENEPY with EXPY is striking. This similarity is a result of the application of our framework, and not an "a priori" construction: in a sense, the economic concepts are self-emerging, with some significant variations with respect to the original EC framework we here reconcile[12,15]. In our view, this similarity represents a possible micro-economically sounded bases for the economic complexity theory, towards which we address future work.

## Methods

**Data**. Import–export data during the year 1995–2017 are extracted from the BACI-CEPII dataset[36], which classifies goods according to the Harmonised System Codes 1992 (HS-1992) at the 6-digits level. To allow comparability with previously published results, we downscale the classification of traded goods to the 4-digits level. Our data include all the countries whose export share is worth at least $10^{-5}$ of the total flux traded during the year (i.e., the total amount of dollars exported worldwide). This filters the noise arising by small export baskets. The Relative Comparative Advantage procedure is used to construct the incidence binary matrix $\mathbf{M}$, setting the threshold of RCA to 1 in line with the economic complexity framework[12]. RCA weights how much a product $p$ counts within the export basket of the country $c$. This fraction is weighted by the ratio of the total monetary flux globally generated by the same product $p$, and the total monetary flux of all products traded worldwide during the reference year. In formulas,

$$RCA_{cp} = \frac{\frac{D_{cp}}{\sum_p D_{cp}}}{\frac{\sum_c D_{cp}}{\sum_{cp} D_{cp}}}, \tag{6}$$

where $D_{cp}$ is the return in dollars of a country $c$ exports through the product $p$. The input matrix $\mathbf{M}$ is given by $M_{cp} = 1$ if $RCA_{cp} \geq 1$, and 0 otherwise[25]. In this work we

also consider the direct use of $RCA_{cp}$ as the input matrix for the computation of the metrics (this implies setting $RCA_{cp} = M_{cp}$), whose results are shown in Supplementary Fig. 10.

**MR metrics**. The equations for the computation of the EC metrics according to the MR approach[12,57] are

$$\begin{cases} ECI_c = \frac{1}{k_c} \sum_p M_{cp} PCI_p, \\ PCI_p = \frac{1}{k_p} \sum_c M_{cp} ECI_c. \end{cases} \tag{7}$$

They can be mapped in our general framework by using

$$\begin{cases} X_c^A = ECI_c \sqrt{k_c}, \\ Y_p^A = PCI_p \sqrt{k_p}, \\ W_{cp}^A = M_{cp}/\sqrt{k_c k_p}. \end{cases} \tag{8}$$

The resulting matrix $W_{cp}^A$ provides with the following symmetric proximity matrices

$$N_{cc^*}^A = \sum_p \frac{M_{cp} M_{c^*p}}{\sqrt{k_c}\sqrt{k_{c^*}} k_p}, \tag{9}$$

for countries, and

$$G_{pp^*}^A = \sum_c \frac{M_{cp} M_{cp^*}}{k_c \sqrt{k_p}\sqrt{k_{p^*}}}, \tag{10}$$

for products. We stress that the matrices $\mathbf{N}^A$ and $\mathbf{G}^A$ are symmetric ones thanks to the presence of the square roots of the degrees $k_c$ and $k_p$, respectively, for which they differ from the corresponding asymmetric matrices that one would obtain directly from the original MR formulation.

Within our framework, the eigenvectors of the two matrices $\mathbf{N}^A$ and $\mathbf{G}^A$ associated to the largest eigenvalue $\lambda_1 = 1$ are $X_{c,1} = \sqrt{k_c}$ and $Y_{p,1} = \sqrt{k_p}$, from which the unitary eigenvectors of the MR framework are recovered through Eq. (8). The second eigenvectors $X_{c,2}$ and $Y_{p,2}$ of the matrices provide the $ECI_c$ and $PCI_p$ solutions using Eq. (8) instead.

**FC metrics**. The non-linear FC algorithm defines the values of complexity, Fitness, $F_c$, for countries and Quality, $Q_p$, for products as[15]

$$\begin{cases} \widetilde{F}_c^{(n+1)} = \sum_p M_{cp} Q_p^{(n)}, & F_c^{(n+1)} = \frac{\widetilde{F}_c^{(n+1)}}{\left(\sum_c \widetilde{F}_c^{(n+1)}\right)/C}; \\ \widetilde{Q}_p^{(n+1)} = \frac{1}{\sum_c M_{cp}\frac{1}{F_c^{(n)}}}, & Q_p^{(n+1)} = \frac{\widetilde{Q}_p^{(n+1)}}{\left(\sum_p \widetilde{Q}_p^{(n+1)}\right)/P}; \end{cases} \tag{11}$$

where $C$ and $P$ are the number of exporting countries and exported products, respectively. In Eq. (11), $\widetilde{F}_c^{(n+1)}$ and $\widetilde{Q}_p^{(n+1)}$ are the intermediate values of $F_c^{(n+1)}$ and $Q_p^{(n+1)}$ obtained at each iteration $(n+1)$[15]. At each step, the intermediate values are normalised by their algebraic means, in this way providing the final values $F_c^{(n+1)}$ and $Q_p^{(n+1)}$. The normalisation is required for the stabilisation of the non-linear map in Eq. (11)[35].

The system in Eq. (11) can be written in closed and non-iterative form as

$$\begin{cases} F_c = c_F \sum_p M_{cp} Q_p, \\ Q_p = c_Q \frac{1}{\sum_c M_{cp}\frac{1}{F_c}}, \end{cases} \tag{12}$$

in which we have embedded the normalisation procedure by introducing the parameters $c_F$ and $c_Q$, namely $c_F = \frac{C}{\sum_p Q_p k_p}$ and $c_Q = \frac{\sum_p Q_p k_p}{P}$. Equation (11) can be seen as the simplest numerical solution of Eq. (12).

Equation (12) represents a functional relationship between the vectors of values $F_c$ and $Q_p$ and, in particular, the Quality values can be formally expressed as $Q_p = h(F_1, F_2, \ldots, F_c)$, $c = [1, \ldots, C]$, where $h(F_1, F_2, \ldots, F_c)$ is a non-linear function of the $C -$ Fitness values. In order to map the FC algorithm onto the linear $X_c$–$Y_p$ framework, we linearise the function $h(F_c)$ using the Taylor's series and expanding the function around the value $F_c = k_c$, which is known to dominate the information contained in $F_c$[19]. Moreover, $k_c$ is the first result of the map at iteration $n = 1$. The Taylor's expansion provides a linear expression to evaluate the Quality of the products, namely

$$\begin{cases} F_c \simeq c_F \sum_p M_{cp} Q_p, \\ Q_p \simeq \frac{c_Q}{(k'_p)^2} \sum_c \frac{M_{cp} F_c}{k_c^2}, \end{cases} \tag{13}$$

where $k'_p = \sum_c M_{cp}/k_c$. Notice that the system in Eq. (13) is an eigen-problem, thus it can be solved without the use of iterative algorithms. This avoids the convergence problem which is known to affect the system in Eq. (11)[29], due to the hyperbolic

nature of the second equation[35]. As stated in the main text, the linearisation of the original definition of $F_c$ and $Q_p$ only biases the results for <0.5% (on average in time according to Pearson's correlation coefficient, Supplementary Figs. 2 and 7).

Taking the linearised equations in Eq. (13) as the starting point, the mapping of FC metrics within our framework is given as

$$\begin{cases} X_c^B = F_c/k_c, \\ Y_p^B = Q_p \cdot k_p', \\ W_{cp}^B = M_{cp}/(k_c k_p'), \end{cases} \quad (14)$$

where we neglect the rescaling factors $c_F$ and $c_Q$, since their roles of stabilising the numerical values is not anymore required due to linearity, thus reducing the number of unknowns in the system.

The resulting matrix $W_{cp}^B = M_{cp}/k_c k_p'$ provides with the following symmetric proximity matrices

$$N_{cc^*}^B = \sum_p \frac{M_{cp} M_{c^* p}}{k_c k_{c^*} (k_p')^2}, \quad (15)$$

for countries, and

$$G_{pp^*}^B = \sum_c \frac{M_{cp} M_{cp^*}}{k_c^2 k_p' k_{p^*}'}, \quad (16)$$

for products. The linearised values for Fitness and Quality are recovered from the eigenvectors of the proximity matrices associated to the largest eigenvalue $\lambda_1$, from which holds $X_{c,1} = F_c/k_c$ and $Y_{p,1} = Q_p k_p'$.

Notice that the computation of the GENEPY index entails interpreting the matrices $\mathbf{N}$ and $\mathbf{G}$ as proximity matrices, thus setting their diagonal elements to same constant values: we here set $N_{cc}^B = G_{pp}^B = 0$. Even when the matrices $\mathbf{N}$ and $\mathbf{G}$ are interpreted as proximity matrices (i.e., their diagonal is set to zero), very good correlations are obtained between linearly and non-linearly computed values (Supplementary Figs. 2 and 7).

**Relation to EXPY metric**. It is easy to verify the similarity of the relation in Eq. (3) (with the elements $N_{cc^*}$ as given in Eq. (15)) to compute the $X_c$ values with the expression to compute the productivity of a country according to the EXPY[2]. In fact, by recalling the weighted incidence matrix of the export volumes in dollars, $D_{cp}$, and the strengths of countries and products such that:

$$k_c = \sum_p D_{cp}, \qquad k_p = \sum_c D_{cp}, \qquad k_p' = \sum_c \frac{D_{cp}}{k_c};$$

one has that the productivity level of a product, named PRODY, is given as

$$PRODY_p = \sum_c \frac{D_{cp}}{k_c k_p'} R_c,$$

being $R_c$ the GDP per capita of the country $c$. EXPY, as a function of the PRODY, is computed as

$$\begin{aligned} EXPY_c &= \sum_p \frac{D_{cp}}{k_c} PRODY_p = \sum_p \frac{D_{cp}}{k_c} \sum_{c^*} \frac{D_{c^* p}}{k_{c^*} k_p'} R_c \\ &= \sum_{c^*} \sum_p \frac{D_{cp} D_{c^* p}}{k_c k_{c^*} k_p'} R_c. \end{aligned} \quad (17)$$

EXPY mainly differs from the GENEPY approach, and thus the linearised FC one, – apart from a rescaling factor $k_p'$ (see Eq. (15)) – for the embedding of the exogenous information on the GDP per capita which replaces the country–country relation in Eq. (3).

**The GENEPY index**. The description on how the GENEPY index is derived from the eigenvectors of the proximity matrices is here exemplified for countries, and the same procedure applies for the index as referred to products. In fact, to obtain the GENEPY index for products, it is sufficient to replace in the following the terms $X_{c,1}$, $X_{c,2}$ and $\mathbf{N}$ with $Y_{p,1}$, $Y_{p,2}$ and $\mathbf{G}$, respectively.

The GENEPY index for countries combines the eigenvectors corresponding to the two largest eigenvalues of the symmetric proximity matrix $\mathbf{N}$ (see Eq. (15)). The manner the information obtained from the two eigenvectors is squeezed into the unique measure in Eq. (5) finds its roots in the recast of the network centrality problem into an estimation exercise. The main steps of this procedure follow, and we refer the readers to the original work[28] for a more detailed explanation.

The matrix $\mathbf{N}$ describes the weighted adjacency matrix of the undirected network whose nodes are the countries and edges the similarities among them. The eigenvectors of this matrix represent centrality measures of the nodes. Our aim is to use the eigenvectors to least-square estimate the matrix $\mathbf{N}$. Firstly, we introduce a centrality-dependent estimator function $\zeta$. In the case of the eigenvector centrality, such function linearly depends on the eigenvectors $X_{c,1}$ and $X_{c,2}$, corresponding to the two largest eigenvalues $\lambda_1$ and $\lambda_2$ of the matrix $\mathbf{N}$[27,28,58]. In formulas

$$\zeta(\lambda_i, X_{c,i}, \mathbf{X}_{s,i}) = \sum_{i=1}^2 \lambda_i X_{c,i} X_{s,i}, \quad (18)$$

where $i = [1, 2]$ and $c$ and $s$ run in the range $[1, C]$, being $C$ the number of countries in the matrix. The function $\zeta$ minimises the squared errors between the matrix elements and the corresponding estimates; namely

$$SE = \sum_c^C \sum_s^C \left( N_{cs} - \zeta(\lambda_i, X_{c,i}, X_{s,i}) \right)^2. \quad (19)$$

Secondly, at a fixed $i^*$, each eigenvector $\mathbf{X}_{i^*}$ solves the minimisation problem[28]

$$\frac{\partial SE}{\partial \mathbf{X}_{i^*}} = 0.$$

In this muldimensional setting on eigenvector centrality, the ranking of the network' nodes (i.e., the countries) is given by the adoption, from the commonality analysis, of the concept of the unique contribution of the $X_{c,i}$ variables. The unique contribution is defined as the drop in the coefficient of determination $R^2$ induced by excluding the variables $X_{c,i}$ ($i = [1, 2]$) considered in the estimator function $\zeta$, in Eq. (18), from the estimation procedure[28]. The core concept of the unique contribution is that, the larger the drop, the larger is the contribution of the $c$-th values in the reconstruction of the matrix $\mathbf{N}$ and, in this application, the more central the $c$-th node is in the network topology under analysis. Hence, according to this approach, we define the GENeralised Economic comPlexitY index (GENEPY) for the country $c$ as the unique contribution of its complexity values $X_{c,i}$ as computed by the formula given in Eq. (5).

**Reporting summary**. Further information on research design is available in the Nature Research Reporting Summary linked to this article.

## Data availability
The trade data supporting the findings of this study are available upon request from the BACI-CEPII database (Gaulier and Zignago[36]). Downloads may require paid subscription.

The GDP PPP and population data used in this work are provided by the World Bank and publicly and freely available at [https://data.worldbank.org/].

The data on the coordinates of countries are provided by the Portland State University and publicly and freely available at [https://www.pdx.edu/econ/country-geography-data].

The pollinators-plants networks are freely available at [www.web-of-life.es].

The results of the GENEPY index for countries during the period of analysis are publicly and freely available at [https://zenodo.org/record/3876721]. Other results are available from the authors upon request.

## Code availability
The code for the computation of the GENEPY index is publicly and freely available at [https://zenodo.org/record/3876721].

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

## Acknowledgements
We acknowledge ERC funding from the CWASI project (ERC-2014-CoG, project 647473).

## Author contributions
C.S., G.C., L.R., and F.L. conceived and designed the study. C.S. prepared the data and conducted the experiments. C.S., G.C., L.R., and F.L. analysed the data. CS produced the figures and wrote the manuscript. C.S., G.C., L.R., and F.L. edited the manuscript. All authors reviewed the manuscript.

## Competing interests
The authors declare no competing interests.
