## [Peer Review File · Nature Communications]

Reviewers' comments:

Reviewer #1 (Remarks to the Author):

The paper provides an interesting and potentially impactful contribution to the Economic Complexity (EC) literature. In particular it introduces a unified linearized framework to reconcile the two most popular bipartite centrality measures used in EC, namely ECI/PCI and Fitness/Complexity.

While I appreciate the mathematical formalism, that allows to provide more rigorous formal interpretations of the metrics, I think the paper lacks in providing more economical interpretations and validations of the new metrics introduced.

The ECI and Fitness formulations were proposed by their authors on the basis of some heuristic reasoning, related to a tripartite countries-capabilities-products network and the 'Building Blocks' combinatorics. In this sense, even if without any formal justification, there is an intuition about what the metrics are trying to capture. While in this work the formalism is certainly better defined, I think the intuitions are much harder to grasp, and there is no explicit validation that suggests that this metric is preferable for any economic task, other than the better formalism itself.

Therefore I would suggest addressing these points to improve the paper:

1. line 24: to be more precise, MR has been introduced in [12] without any reference to the algebraic formulation. The algebraic formulation has been introduced in [A] and further expanded in [B], much earlier than [14] and [16]. Since [A, B] cast the MR on the same linear algebra framework on which also this paper builds and expands, I think it would be fair to cite and contextualize these works.

2. line 74: it is very interesting to find such an high correlation, however this raises a few questions/comments that can be addressed to better contextualize the results:

- It is important to notice that one of the main features of Fitness is to be a sum (not an average) over the products. Here the authors divide the Fitness by k_c , therefore transforming it into an average. This passage is non trivial and can have a big impact on the rankings: the authors should comment on this. In what context and why is this desirable? Isn't diversification itself an important point?

- Are the deviations from this correlation informative of something? What countries are the biggest outliers? Can this help explaining the relation between Fitness and its linearized version?

- Is this correlation a specific feature of the countries-products bipartite network or is it found also in other contexts? E.g.: would such high correlation be found on sparser/denser bipartite networks? On more/less nested networks? In random networks? In bipartite networks arising in other domains (ecology, biology, technology)? I don't expect the authors to systematically explore these questions, as that would be an entirely new paper. However I would suggest to run some test at least on perturbations of the countries-products network, to provide a better understanding of the similarity and differences between Fitness and this linearized counterpart.

- Fitness/Complexity is known to give a close-to-perfect nested ordering of the rows and columns of the bipartite matrices [C, D]. One way to look at this is to notice that it allows to very efficiently 'pack' all the non-zero elements of such matrices in a 'triangle'. Is the 'packing' induced by this linearized version better, equal or worse than that arising from Fitness? This could be another way to understand in which sense this linearized version approximates Fitness.

3. line 80: 'only within this framework...' I am not sure about what the authors mean with this sentence. There are many other ways to project a bipartite network into a monopartite one that can be interpreted as a proximity network. Can the authors better clarify what they mean?

4. lines 88-89: they are eigenvalues of different order, but also of different matrices. I might agree that it is pointless to compare the two metrics with respect to what they say on the topology of

networks, however these metrics are typically compared on how much they can say about the economic status of countries. In this context is perfectly fine to compare different topological properties in order to see which one carries more information about economies. I think the authors should better clarify this sentence.

5. line 98: isn't the first eigenvector of MR a constant vector?

6. lines 106-108: here all the explanations are left to the methods sections. However, to improve the readability of the paper, I suggest to put here some comments on what is the general goal and some intuition of what is the rationale of this combination. Derivation and more technical comments can be left in the methods or in the references.

7. Fig 1: in panel a it seems to me that there are two different trends, decreasing before $X_{c,1}=0.05$, and then increasing. Does this have some interpretation? Is $X_{c,1}=0.05$ a special point wrt some economic dynamics? Is the correlation between $X_{c,1}$ and $X_{c,2}$ generally expected? Can we say something about the outliers? E.g.: CHN vs SGP?

8. Fig 1: in panel b I would label the outliers and comment. Also I would add a comment on why the residuals seem to be mostly skewed upwards. Does this help answering to point 2.3 of my comments? Along the same lines, I would also comment on the residuals of ECI vs $X_{c,2}$. These are clearly simply due to the choice of how to build the similarity matrix (being the method perfectly identical to ECI): can we gain some interpretation about the validity of this choice?

9. line 152: this and the following comment (Arena) seems to imply that there are 2 clear clusters, but I don't find them so sharply. Is there any theoretical understanding of why $X_{c,2}=0$ should be a transition point? Is it possible to give a clearer understanding of what the $X_{c,2}$ dimension captures and in what sense (in terms of their export basket size and composition) high $X_{c,2}$ countries are developed? This can be said also in contrast with having high $X_{c,1}$ but not $X_{c,2}$ or vice-versa (see point 7)

10. lines 225-227: "As such... countries." I don't agree with this statement. All the economic complexity metrics are essentially more or less sophisticated ways of measuring topological features of the locations-activities networks (see minor comment 1 below). While mathematically these metrics can have better or worse properties, it is only by comparing them against other sources of information that one is able to understand the actual value of the metrics. If we know these metrics do we know something more about the state of the system? Are we able to make better predictions? Or to do them with less assumptions? Less or noisier data?

11. Following my previous comment: can the authors give some ideas of how does GENEPY compare to some standard macroeconomic features of countries (e.g. GDP)? Is there a reason to believe that GENEPY is actually carrying more information about these than Fitness or ECI?

Some minor comments:

1. In the abstract the authors state that Economic Complexity is about the diversity of product exports. While this was true for the original works, nowadays the field has extended much beyond, and the more general subject of EC can be better described as the diversity of (human) activities. Especially in the field of economics EC has been often confused as a framework to study global trade, and I think it is important to not further fuel this misunderstanding.

2. line 179: does it really make sense to multiply an intensive complexity measure by the population of the country? What does that mean?

To conclude, I have found this contribution very interesting at least in principle. However I think the manuscript in its present form does not allow to evaluate if this contribution adds substantially to the field, or if it just provides a neat mathematical framework but no additional forecasting or explanatory power. My comments are mostly in the direction of getting hints about this point (actually proving a better forecasting power would probably be beyond the scope of this paper). I don't really expect that

the authors perform all the exercises that I propose, I just intend them as a stimulus for the authors to find ways to better show the strength of their contribution.

References

[A] A network analysis of countries' export flows: firm grounds for the building blocks of the economy
G Caldarelli, M Cristelli, A Gabrielli, L Pietronero, A Scala, A Tacchella. PloS one 7 (10), e47278

[B] Measuring the intangibles: A metrics for the economic complexity of countries and products M
Cristelli, A Gabrielli, A Tacchella, G Caldarelli, L Pietronero. PloS one 8 (8), e70726

[C] Nestedness in complex networks: Observation, emergence, and implications, MS Mariani, ZM Ren,
J Bascompte, CJ Tessone. Physics Reports 813, 1-90

[D] Ranking species in mutualistic networks. V Domínguez-García, MA Muñoz. Scientific Reports (2015)
10.1038/srep08182 5, 8182

Andrea Tacchella

Reviewer #2 (Remarks to the Author):

The paper provides an interesting and novel insight on a series of previously proposed metrics to assess in a synthetic way the capability stock of countries. It surely contributes to the previous literature in a constructive way providing a general framework to better understand the differences of existing metrics and how to, potentially, improve them, especially the Fitness.

The paper is well written, it is extremely neat and pleasant to read.

I would have only two major remarks:

i) figure 2 and related analysis in the text: differently from the first part, I find the analysis a bit weak and the three regimes sound a bit anecdotal. The authors should make this part more quantitative. As a simple (and minimum) step in this direction, they should somehow smooth and aggregate the dynamics in their plane to show the emerging flow as did in [1,2,3] for instance and try to show the three regimes. The analysis would be even more robust and complete, if the authors would try an analysis in the spirit of what did in [4] where with a simple Granger causality analysis it is shown that there are different regimes of interaction between two variables in the perspective of a dynamical system - the Granger test is just a suggestion, it can be any other techniques which is considered by the authors appropriate to quantify the effects in this plane. We should expect that in the left bottom corner, X1 and X2 to be essentially unrelated - causality speaking -, in the middle part, they should be related and perhaps one is leading the other - this would be an interesting question to address -, top right part, I do not know what to expect a priori.

ii) the authors often state that the high correlation of two metrics is somehow implicitly a proof of the fact that they carry similar information. For instance, it is used to justify that the non-linear specification of the fitness is likely not needed. While, generally speaking, I essentially agree with the content of the paper, I strongly disagree with the authors on this specific point. A priori, we do not know in which part of the variables is stored the informative content, two variables might be 99% correlated but the difference in their forecasting power can be paradoxically huge and all concentrated in the residual part. Let us consider two variables $Z1 = a X + (1-a) Y1$ and $Z2 = a X + (1-a) Y2$, and $Y1$ is orthogonal to $Y2$ and $Y1$ is signal while X and $Y2$ are noise. If 'a' is close to 1, the two variables

will be highly correlated but only Z2 will be useful to forecast/explain something, the signal-to-noise ratio will be surely weak, but only Z2 will have signal despite the high correlation with Z1. This is for instance the case of ECI and Fitness, they are correlated, but once used in practice to forecast or explain economic dynamics, the discrepancies distinguish the two variables much more than what the correlation would suggest. I would suggest to either downgrade the argument which is leveraged several times or, more involved, to show that the linearized fitness and the non-linear have a similar predictive/explanatory power (see for instance last suggestion). Provided the correlation argument only, the conclusion cannot be drawn and the statement should be only descriptive. As a side comment, the authors should also show that the results of the correlation does not depend on the specific estimator, if they use Spearman or Kendall's Tau correlation, do the results change?

A minor comment concerns figure 3, the one of the three different barycenters. The fact that the GDP barycenter is still far from the one provided by the GENEPI index is likely the sign that the potential of economic growth of Asian countries is still strong (even for China) as observed in [3].

As a general suggestion, likely for the next paper - it would interesting to test GENEPI index in a framework like the one proposed in [3] because this would be the true benchmark to see if GENEPI carries more information than Fitness and ECI and if the linearized part of the Fitness X1 is really carrying the same information of the non-linear counterpart.

[1] Pugliese et al. Economic Complexity as a Determinant of the Industrialization of Countries, 2017

[2] Cristelli et al, The heterogeneous dynamics of economic complexity, 2015

[3] Cristelli et al., On the predictability of growth, 2017

[4] Cristelli et al., The Virtuous Interplay of Infrastructure Development and the Complexity of Nations, 2018

Matthieu Cristelli

Reviewers' comments:

Reviewer #1 (Remarks to the Author):

The paper provides an interesting and potentially impactful contribution to the Economic Complexity (EC) literature. In particular it introduces a unified linearized framework to reconcile the two most popular bipartite centrality measures used in EC, namely ECI/PCI and Fitness/Complexity.

While I appreciate the mathematical formalism, that allows to provide more rigorous formal interpretations of the metrics, I think the paper lacks in providing more economical interpretations and validations of the new metrics introduced.

The ECI and Fitness formulations were proposed by their authors on the basis of some heuristic reasoning, related to a tripartite countries-capabilities-products network and the 'Building Blocks' combinatorics. In this sense, even if without any formal justification, there is an intuition about what the metrics are trying to capture. While in this work the formalism is certainly better defined, I think the intuitions are much harder to grasp, and there is no explicit validation that suggests that this metric is preferable for any economic task, other than the better formalism itself.

We thank this reviewer for the positive feedback about our work. Generally, we agree with him about the grounding reasoning of the present work. In fact, the aim of this work is not to present a new indicator, rather to show how the two existing metrics, ECI and Fitness, can be joined in a unique, neat framework, thus providing a tool to exploit the potential (and the economic significance and validation) of both metrics in assessing the hidden capabilities of countries. It follows that our framework inherits the same intuitions/assumptions both methodologies (firstly FC, secondly ECI) have introduced about the way countries manage and distribute their capabilities in their export baskets, from which the complexity of products is determined. This is implicit in the way we compute the GENEPEY indices for countries and products, which have as input the matrices from the linear mapping of the FC algorithm (i.e., Eqs (11) – (14)). Moreover, to combine ECI and FC (as the result of the neat framework we here present, not an “a priori” construction) resolves the old (and sometimes harsh) debate on which index to use; a debate that in our view has weakened the application of Economic Complexity in the field of economic studies.

We have made this explicit by editing line 103 of the revised manuscript as follows:

“...FC carries more information than MR. The grounding hypotheses about the hidden capabilities of countries – and on how these can be deduced looking at the export baskets of countries upon which the EC algorithms are built – are preserved in our framework. From here on...”

And at line 146:

“...Eq (15), respectively. Being the GENEPEY framework grounded on both existing indicators of economic complexity (the FC and the MR algorithms), it inherits the intuitions and rationales upon which these two metrics are built: the capabilities of countries to export diversely complex goods are hidden within the bipartite network of countries and exports, under which they combine to maximize the complexity of the goods.”

For what concerns validation, we embrace the idea of economic complexity as a driver for growth [a, d]. Any formal validation would require defining an objective function (e.g., predictive power of economic growth) and verifying if using the GENEPEY as an additional independent variable allows one to get closer to the selected objective (e.g., to improve the predictive power). However, the validation problem is somewhat ill-posed: the standard use of the GDP to measure growth, for example, is contradictory within a framework – the economic complexity one – aiming to overcome a simplistic, one-dimensional, view of the economic dynamics. One would be tempted to use the economic complexity measures to uniquely represent economic growth, but this would induce a logical loop in the system, where the validation is based on the same variable

subject to validation. We therefore deliberately decided not to enter the validation arena in this work, trusting the validation efforts performed by others on the Fitness and ECI measures [a,b,e], and taking advantage of their results to also support our multidimensional metrics, whose components are in fact strictly related to Fitness and ECI.

Lastly, we would like to highlight, once more, that – from an economical point of view – this work could pave the way to the microeconomics foundation of the Economic Complexity field, due to the similarities of the formalisms among the GENEPY and the EXPY [c]. Once again we underline that this similarity is a result of the application of our framework, and not an “a priori” construction: in a sense, the economical concepts are self-emerging, with some significant variations with respect to the original economic complexity works, from the reformulation of the intuitions of MR and FC within a neater mathematical framework. We argue that this aspect is a fundamental one for the improvement of data-science based economics. Concerning with this comment, some sentences have been added at lines 265 and on:

“Moreover, in the FC algorithm the Quality of a product is mainly determined by the least fit country exporting it, a crucial property accomplished by the non-linearity of the FC approach. In our linear framework, this property is maintained through the term $k'_p = \sum_c M_{cp}/k_c$, occurring in $W_{cp} = M_{cp}/k_c k'_p$. This term in fact represents the degree of a product corrected by how easily it is found within the network. Its inverse $1/k'_p$ is an anti-centrality score for the product, determining how limited is its presence within the producers' baskets and thus suggesting the need for higher productive knowledge in its production process. Notice that, by substituting the incidence matrix M with the traded monetary values, the term k'_p also recurs in the so-called EXPY rationale by Hausmann et al. [c] Based on a decision-making model of firms' investment choices, the Authors in [c] defined an index of economic growth potential of countries, assessed through the required productive level of the exported products, i.e., EXPY. As we show, (see Methods, Eq (17)), the equations to compute X_c in the GENEPY framework are similar to those defining the EXPY scores of countries [c]. Clearly, EXPY has been defined from a different deductive rationale, which considers the trade as described by the weighted incidence matrix of the monetary fluxes (thus providing different input information) and embeds exogenous information such as the GDP per capita. Notwithstanding these differences, the formal similarity of GENEPY with EXPY is striking. This similarity is a result of the application of our framework, and not an “a priori” construction: in a sense, the economic concepts are self-emerging, with some significant variations with respect to the original EC framework we here reconcile [a,b]. In our view, this similarity represents a possible micro-economically sounded bases for the economic complexity theory, toward which we address future work.”

[a] Tacchella, A., Cristelli, M., Caldarelli, G., Gabrielli, A., & Pietronero, L. (2012). A new metrics for countries' fitness and products' complexity. *Scientific reports*, 2, 723.

[b] Hidalgo, C. A., & Hausmann, R. (2009). The building blocks of economic complexity. *Proceedings of the national academy of sciences*, 106(26), 10570-10575.

[c] Hausmann, R., Hwang, J., & Rodrik, D. (2007). What you export matters. *Journal of economic growth*, 12(1), 1-25.

[d] Hausmann, R., Hidalgo, C. A., Bustos, S., Coscia, M., Simoes, A., & Yildirim, M. A. (2014). *The atlas of economic complexity: Mapping paths to prosperity*. Mit Press.

[e] Tacchella, A., Mazzilli, D. & Pietronero, L. (2018) A dynamical systems approach to gross domestic product forecasting. *Nature Phys* 14, 861–865.

Therefore I would suggest addressing these points to improve the paper:

1. line 24: to be more precise, MR has been introduced in [12] without any reference to the algebraic formulation. The algebraic formulation has been introduced in [A] and further expanded in [B], much earlier than [14] and [16]. Since [A, B] cast the MR on the same linear algebra framework on which also this paper builds and expands, I think it would be fair to cite and contextualize these works.

We thank the reviewer for raising this point and for the useful readings suggested. We have changed line 24 as follows:

“The equations defining the two averages are coupled to obtain the Economic Complexity Index, ECI, and the Product Complexity Index, PCI, [11,12], which have been shown to be the result of a linear algebra exercise [A,B,11].”

2. line 74: it is very interesting to find such an high correlation, however this raises a few questions/comments that can be addressed to better contextualize the results:

- It is important to notice that one of the main features of Fitness is to be a sum (not an average) over the products. Here the authors divide the Fitness by k_c , therefore transforming it into an average. This passage is non trivial and can have a big impact on the rankings: the authors should comment on this. In what context and why is this desirable? Isn't diversification itself an important point?

We thank the reviewer for this point. A response pertains with a possible misunderstanding: $X_{c,1}$ and F_c/k_c carry approximately the same information (correlation coefficient larger than 0.98), while $X_{c,1}$ and F_c do not. In fact, the mentioned sentence at line 74 refers to a figure in the SI included to “validate” the linearization procedure, where the comparison analysis is performed between variables sharing the same meaning (i.e., $X_{c,1}$ and F_c/k_c , or, analogously $X_{c,1} * k_c$ and F_c). We clarified this concept at lines 75:

*“... See Methods, Eqs (11) – (13)). Surprisingly enough, comparing the terms $X_{c,1}^B$ and F_c/k_c , or $X_{c,1}^B * k_c$ and F_c , for the Fitness values – analogously $Y_{p,1}^B$ and $Q_p k'_p$ (or $Y_{p,1}^B/k'_p$ and Q_p) for the Quality values – this linearization preserves >98% of the information (independently of the kind of indicator of correlation chosen, Figure S2), thus questioning ...”*

The rankings obtained with $X_{c,1}$ and F_c are indeed different ones, and in this sense this reviewer has a good point in arguing that transforming the sum (F_c) into an average ($X_{c,1}$) might imply that diversification is lost from our metrics, with disputable economic implications. However, this is not the case: $X_{c,1}$ maintains a very high correlation with diversification (as Figure 1 shows, >0.85 correlation coefficients – both in Spearman and in Pearson formulations). This might appear surprising, because $X_{c,1}$ is obtained by dividing F_c by k_c , and one could expect taking the ratio kills correlation: however, this would be the case only if F_c and k_c were perfectly linearly related, which of course is not the case. Since this relation is super-linear (because more sophisticated products are produced by countries with highly diversified baskets), $X_{c,1}$ preserves information on diversification. We have included some discussion on this very relevant issue after the comment added at line 146 of the revised manuscript:

“...Eq (15)), respectively. Being the GENEPI framework grounded on both existing indicators of economic complexity (the FC and the MR algorithms) it inherits the intuitions and rationales upon which these two metrics are built: the capabilities of countries to export diversely complex goods are hidden within the bipartite network of countries and exports, under which they combine to maximize the complexity of the goods. Also, since $X_{c,1}$ maintains a very high correlation with k_c (see SI, Figure S6), our framework preserves the information on diversification, which is a relevant one to understand how export capabilities are exploited by countries.”

Figure 1: Correlation between $X_{c,1}$ and k_c . In panel (a), the scatter plot of the values from year 2017. In panel (b), the values of the correlation coefficients between the two vectors during time. The correlation of the Pearson's kind is in green, while the Spearman's one in orange.

- Are the deviations from this correlation informative of something? What countries are the biggest outliers? Can this help explaining the relation between Fitness and its linearized version?

We thank the reviewer for raising this point. The mapping in Eqs (14) solves the linearized version of the FC algorithm in Eqs (13) by defining the symmetric matrices \mathbf{N} and \mathbf{G} (Eqs (15) – (16)). As we explain, the GENEPEY arise from the eigenvectors of the matrix \mathbf{N} , this one being interpreted as proximity matrix. This interpretation entails setting the elements of the matrix as:

$$\begin{cases} N_{cc^*} = \sum_p \frac{M_{cp}M_{c^*p}}{k_c k_{c^*} (k_p')^2}, c \neq c^* \\ N_{cc^*} = 0, c = c^*; \end{cases}$$

i.e., the diagonal values are set to zero (any other uniform value would lead to the same eigen-result; we lean toward the use of zero because it allows one to delete, in network jargon, the unnecessary information about the self-loops). If one left the diagonal values as resulting from the linearization and mapping procedure, such values would be different among different countries, and this would corrupt the interpretation of \mathbf{N} as proximity matrix (in fact each country has perfect similarity to itself and, therefore, all diagonal entries have to be equal).

However, to use the mapping in Eqs (14) without modifying the diagonal values allows one to almost-perfectly recover the outcomes of the non-linear iterative algorithm with 99.9% accuracy, as shown here in Fig 2 of this reply. Interestingly, this demonstrates that the linearization procedure does not induce any loss of information, since even the very small deviations from the 1:1 line shown in Figure 2, panel a (the so-called outliers) disappear when the equation $N_{cc^*} = \sum_p \frac{M_{cp}M_{c^*p}}{k_c k_{c^*} (k_p')^2}$ is used also for $c = c^*$. However, this would imply inflating the F_c (or $X_{c,1}$) values for countries with large self-interactions, which would induce a bias in the results. Again, interesting economic issues emerge from the application of our framework.

Figure 2: Scatter plots of the eigenvector $X_{c,1}$ and F_c/k_c . On the left, the eigenvector $X_{c,1}$ belongs to the matrix \mathbf{N} with diagonal values set to zero. On the right, $X_{c,1}$ is the eigenvector of the matrix \mathbf{N} in which we leave the diagonal values as computed. Data refer to year 2017.

We included comments on this issue on the updated version of the manuscript starting at line 139 of the revised manuscript:

“[...] (Figure S2). The very small deviations from the 1:1 line shown in panel (b) are not induced by the linearisation procedure. In fact, they disappear when the equation $N_{cc^*} = \sum_p \frac{M_{cp}M_{c^*p}}{k_c k_{c^*} (k'_p)^2}$ is used also for $c = c^*$, i.e., when the matrix \mathbf{N} is not interpreted as a proximity matrix (see Methods, Eq (13) and SI, Figure S5). However, this would imply inflating the F_c (or $X_{c,1}$) values for countries with large self-interactions, which, in our opinion, induces an undesired bias in the results. Analogously, a good proxy of the ECI values is obtained [...]. In this case, the scatter of the plot is due to the differences in the matrices \mathbf{N}^A and \mathbf{N}^B (see Methods, Eq (9) and Eq (15)), respectively.”

- Is this correlation a specific feature of the countries-products bipartite network or is it found also in other contexts? E.g.: would such high correlation be found on sparser/denser bipartite networks? On more/less nested networks? In random networks? In bipartite networks arising in other domains (ecology, biology, technology)? I don't expect the authors to systematically explore these questions, as that would be an entirely new paper. However I would suggest to run some test at least on perturbations of the countries-products network, to provide a better understanding of the similarity and differences between Fitness and this linearized counterpart.

- Fitness/Complexity is known to give a close-to-perfect nested ordering of the rows and columns of the bipartite matrices $[C, D]$. One way to look at this is to notice that it allows to very efficiently 'pack' all the non-zero elements of such matrices in a 'triangle'. Is the 'packing' induced by this linearized version better, equal or worse than that arising from Fitness? This could be another way to understand in which sense this linearized version approximates Fitness.

Thanks for these comments. We are aware of the potential the FC algorithm has in minimizing the nestedness temperature of ecological networks [f], and in this field non-linearity has been shown to be an important feature of the algorithms for temperature minimization [g]. As detailed in the Discussion, our work only pertains with the specific structure of the countries-products bipartite network, with no ambition to build up an algorithm with more general ‘data-packing’ potential, where indeed we expect nonlinearity to play a major role. To confirm this statement, we have tested the packaging performance of the linearized form of the FC algorithm, similarly to the comparison showed in [f]. We exemplify the results through the analysis of the pollination networks provided by The Web of Life project, www.web-of-life.es, (network IDs : M_PL_062 and : M_PL_015). The networks describe the pollination phenomena among plants and pollinators. As Figure 3 shows, the non-linear algorithm outperforms the linearized form in the capability of maximizing the nestedness of the matrices for the two pollination networks we have taken for the example. Instead, there are no significant differences between the non-linear and the linear algorithm for maximizing the data-packing of the trade matrix, confirming that the feature of linearity only pertains with the countries-products bipartite network.

Figure 3: Nestedness capability – comparison among the performances of the non-linear FC algorithm (panels a) and its linearized form, (panels b). The left panels refer to the countries-products bipartite network during 2017. The central panels refer to the network of pollination in Carlinville, Illinois, USA (network ID: M_PL_062); on the right the one referring to the pollination in Daphni, Athens, Greece (network ID: M_PL_015). Data for the pollination networks are freely available at www.web-of-life.es.

We specify that the good performances of the linearized algorithm cannot be extended to other fields beyond economic complexity. We enforced the discussion about this point in the Discussion section, line 255 and on:

“[...], instead. The good performances of the linearised version of the FC algorithm cannot be systematically generalized to other fields beyond economic complexity; in fact, some bipartite systems require a genuine non-linear approach to let their complex and nested nature emerge. However, the good results obtained in the field of EC suggest that there are systems where non-linearity plays a negligible role. We speculate...”

[f] Lin, Jian-Hong, Claudio Tessone, and Manuel Mariani. "Nestedness maximization in complex networks through the fitness-complexity algorithm." *Entropy* 20.10 (2018): 768.

[g] Wu, Rui-Jie, et al. "The mathematics of non-linear metrics for nested networks." *Physica A: Statistical Mechanics and its Applications* 460 (2016): 254-269.

- line 80: ‘only within this framework...’ I am not sure about what the authors mean with this sentence. There are many other ways to project a bipartite network into a monopartite one that can be interpreted as a proximity network. Can the authors better clarify what they mean?

We thank the reviewer for this insight. To the best of our knowledge, the presented mathematical framework is the only one that allows one to merge the two analyses related to the countries-products bipartite network,

i.e., measuring competitiveness and simultaneously, using the same matrix, defining the similarities among countries. In our view, this feature is an important one:

For better clarify this point, we have modified the text at lines 82 - 84 as follows:

“The use of the variables X_c and Y_p allows one to gain neatness in the mathematics, also reflected by the fact that the matrices N and G can be considered as suitable proximity matrices containing information about the similarities among countries and products, respectively. This aspect ...”

4. lines 88-89: they are eigenvalues of different order, but also of different matrices. I might agree that it is pointless to compare the two metrics with respect to what they say on the topology of networks, however these metrics are typically compared on how much they can say about the economic status of countries. In this context is perfectly fine to compare different topological properties in order to see which one carries more information about economies. I think the authors should better clarify this sentence.

We thank the reviewer for pointing this out. In this framework we are showing that, these metrics arise from some structural features of the countries' proximity networks, N^A and N^B , for MR and FC, respectively. Although the matrices N^A and N^B are different, as we show in Figure S3, there exist 88% of correlation among the eigenvectors of the same order of the matrices N^A and N^B . Therefore, even considering the matrix N^B and its corresponding eigenvectors $X_{c,1}^B$ and $X_{c,2}^B$, the information these vectors are bringing about the structural properties of the similarities across the countries are differently relevant. In this sense, we agree with the reviewer that the comparison should only be performed at the analytical level, not the economic one.

We have updated the text at line 89 – and on – as follows:

“... notwithstanding the differences among the matrices N from which these metrics are recovered, the eigenvectors $X_{c,1}^A$ and $X_{c,1}^B$ carry similar information (see Figure S3), as also $X_{c,2}^A$ and $X_{c,2}^B$ (this is also partially true for Y_p , see SI, Figure S6). Therefore, the divergences between Fc and ECI – and corresponding outcomes – shown in Figure S1 should be mainly attributed to the fact that eigenvectors of different order are considered in the two approaches. Hence, the two metrics bring ...”

5. line 98: isn't the first eigenvector of MR a constant vector?

*The eigenvector we are referring to is $X_{c,1}^A = 1 \cdot \sqrt{k_c}$ of the matrix $N_{cc}^A = \sum_p \frac{M_{cp} M_{c^*p}}{\sqrt{k_c} \sqrt{k_{c^*}}}$. The mapping we describe in Eqs (8) – (10) allows us to understand the reason why the first eigenvector of MR is constant: the information that gets “lost” with the unitary eigenvector is exactly the term $\sqrt{k_c}$. The reason for this is the fact that, as also shown in the references the reviewer suggested, MR metrics arise from random walk matrices which have, by definition, unitary principal eigenvector. However, by recasting MR within this framework of symmetric, similarity matrices we re-uncover this feature, adding a further step on the analysis of the results. We have clarified this point at line 100:*

“... (i) the first eigenvector $X_{c,1}^A$ – from which, using Eqs (8), the unitary first eigenvector of MR is recovered – equals $\sqrt{k_c}$...”

6. lines 106-108: here all the explanations are left to the methods sections. However, to improve the readability of the paper, I suggest to put here some comments on what is the general goal and some intuition of what is the rationale of this combination. Derivation and more technical comments can be left in the methods or in the references.

We are thankful to the reviewer for this useful suggestion. More details have been included in the revised version of the manuscript at the same point in the text, now lines 110 – 115:

“... to zero). The rationale to compute the GENEPI index grounds on two key points: (i) to interpret the symmetric squared matrix N as the mathematical description of the weighted topology of an undirected network $[h]$ – such that the countries are the nodes and the similarities between the export baskets are the links connecting them – and, consequently, (ii) to interpret the eigenvectors of N as the (multidimensional) eigenvector centrality of the nodes in the network. Using this approach, the eigenvectors are combined into a unique metrics (the GENEPI one), following a statistically grounded framework where the same eigenvectors are obtained as the result of a least-squares estimation exercise $[i]$.”

$[h]$ Newman, M. E. *Network - An introduction* (Oxford University Press, 2010)

$[i]$ Sciarra, C., Chiarotti, G., Laio, F. & Ridolfi, L. A change of perspective in network centrality. *Sci. Reports* 8 (2018).

7. Fig 1: in panel a it seems to me that there are two different trends, decreasing before $X_{c,1}=0.05$, and then increasing. Does this have some interpretation? Is $X_{c,1}=0.05$ a special point wrt some economic dynamics? Is the correlation between $X_{c,1}$ and $X_{c,2}$ generally expected? Can we say something about the outliers? E.g.: CHN vs SGP?
8. 9. line 152: this and the following comment (Arena) seems to imply that there are 2 clear clusters, but I don't find them so sharply. Is there any theoretical understanding of why $X_{c,2}=0$ should be a transition point? Is it possible to give a clearer understanding of what the $X_{c,2}$ dimension captures and in what sense (in terms of their export basket size and composition) high $X_{c,2}$ countries are developed? This can be said also in contrast with having high $X_{c,1}$ but not $X_{c,2}$ or vice-versa (see point 7)

We thank the reviewer for pointing these observations out. We merge the answers to these two points because they are related.

The knee-shape of the points in the plane $X_{c,1} - X_{c,2}$ is recurrent in all the years of analysis, thus showing the existence of a functional relationship between the two eigenvectors. The reasons of the knee-like shape of this functional relationship are related to linear algebra and network science.

Let us define a functional relationship f between $X_{c,1}$ and $X_{c,2}$ s.t.

$$X_{c,2} = f(X_{c,1}) + \epsilon_c, \quad (\text{Eq 1})$$

where ϵ_c are the errors. We assume the errors to have null expected value, i.e., $E(\epsilon_c) = 0$ and to be orthogonal to $X_{c,1}$, s.t., $\sum_c X_{c,1} \epsilon_c = 0$.

There exist some constraints related to the existence of the eigenvectors of a symmetric matrix, which any functional relationship should respect:

- (i) the eigenvectors corresponding to distinct eigenvalues of a symmetric squared matrix are, by definition, orthogonal and this entails that the inner product of the vectors is zero, i.e., $\sum_c X_{c,1} \cdot X_{c,2} = 0$;
- (ii) for the Perron-Frobenius theorem, the eigenvector corresponding to the largest eigenvalue is strictly positive, s.t. $X_{c,1} \geq 0 \forall c = 1, \dots, C$ (number of countries);
- (iii) we can normalize the eigenvectors such that the 2-norm is unitary, i.e., $\sum_c X_{c,1}^2 = \sum_c X_{c,2}^2 = 1$;

- (iv) if any element of the eigenvector corresponding to the first (largest) eigenvalue λ_1 is zero, the same element is null also within the successive eigenvectors. In fact, the eigen-equation for the matrix \mathbf{N} is:

$$X_{c^*,1}\lambda_1 = \sum_c N_{cc^*}X_{c,1};$$

because of condition (ii), it holds that $X_{c^*,1} = 0$ iff $N_{cc^*} = 0$, i.e., if the matrix has null elements along the column (or row) c^* . Interpreting this result through network science lenses, the node to which the null element of the eigenvector refers is disconnected in the network. Therefore, in the hypothesis of existence of any functional relationship between two eigenvectors as in Eq 1, it must hold $f(0) = 0$.

We now proceed exploring two cases of possible functional relationship for Eq 1.

CASE A: The simplest form of this relation considers f as a linear function, i.e.,

$$X_{c,2} = aX_{c,1} + \epsilon_c \quad (\text{Eq 2})$$

By imposing the orthogonality condition (i) to Eq 2 one obtains:

$$\sum_c X_{c,1}X_{c,2} = \sum_c X_{c,1}(aX_{c,1} + \epsilon_c) = a \sum_c X_{c,1}^2 + \sum_c X_{c,1}\epsilon_c = 0$$

Since the errors are orthogonal to $X_{c,1}$ and the 2-norm of the vector is unitary for condition (iii), the solution is $a = 0$, which entails no functional relationship exists between $X_{c,1}$ and $X_{c,2}$.

CASE B: We consider the function to be polynomial of the second order, namely:

$$X_{c,2} = aX_{c,1} + bX_{c,1}^2 + \epsilon_c \quad (\text{Eq 3})$$

Again, by applying the orthogonality condition (i), one has:

$$\sum_c X_{c,1}X_{c,2} = \sum_c X_{c,1}(aX_{c,1} + bX_{c,1}^2 + \epsilon_c) = a \sum_c X_{c,1}^2 + b \sum_c X_{c,1}^3 + \sum_c X_{c,1}\epsilon_c = 0$$

which leads to

$$a + b \sum_c X_{c,1}^3 = 0 \quad (\text{Eq 4})$$

Because of condition (ii), the term $\sum_c X_{c,1}^3$ is strictly positive and in order to respect Eq 4, the values of the parameter a and b should have different signs, thus justifying the existence of the knee-like shape. In particular, the upward belly of the relation is given for negative values of the parameter a and positive values of b . In this sense, the minimum point depends on the parameters.

We can read this result through the meaning of the matrix and its eigenvectors in the context of network theory. In fact, in this case the eigenvectors of the matrix describe the structural properties of the network $[n]$ and are related to the similarity of the network among the countries. In fact, the shape of the matrix \mathbf{N} (Figure S4), represents a connected network in which a stronger connected component can be spotted, constituted by the top-GENEPY countries, while weaker connections characterize the countries at the periphery. In this weak connection component, as shown in [1], the correlation between the two eigenvectors is positive. Also, as stated by the authors in [1], the mutual signs of the elements of the eigenvectors corresponding to the two largest eigenvalues – whether these are positive or negative in the second eigenvector – acquires a meaning, thus justifying the presence of three areas (or groups) in which the points can stand:

- i) both values $X_{c,1}$ and $X_{c,2}$ are low: these nodes belong to the weaker component and they have no important connections with the strongest connected component;
- ii) both values $X_{c,1}$ and $X_{c,2}$ are high: these nodes belong to the strongest and more connected core of the network, thus defining the area in which these points (countries) are competitors, also in the sense of collecting most of the links in terms of similarities;
- iii) Low values of $X_{c,1}$, high values of $X_{c,2}$ or viceversa: this situation identifies the presence of some “outliers” of the core and the periphery components. These nodes connect the stronger and the weaker components and have a role in bridging the gaps across the network. We identify these nodes as able to jump from one group to another.

We added some comments on this part at line 163 and on, and included this analysis in the SI, S1.1:

“... One recognizes that also the ensemble of the trajectories is knee-shaped: in fact, in each year of analysis the positions of countries in the plane $X_{c,1} - X_{c,2}$ configures in a knee-like shape as shown in Figure 1 for the year 2017. The presence of this shape is related to linear algebra and network science (see SI, S1.1).”

As regards the ability of $X_{c,2}$ to cluster countries, the authors in [m] have proved that ECI perfectly solves a spectral clustering exercise in a network, with the sign of the eigenvector discerning to which cluster the nodes belong. The matrix we use to construct the GENEPY does not coincide with the same matrix from the Method of Reflection; however, the good correlation we have shown in Figure 1c suggests that the sign of $X_{c,2}$ substantially preserves the clustering information of ECI.

Some comments about this point have been added in the revised manuscript at line 129:

“... cluster countries according to the similarities in their export baskets. In fact, the strict nexus between $X_{c,2}$ and ECI recalls the results provided in [m], where the Authors proved that ECI perfectly solves a spectral clustering algorithm. Interpreting...”

Lastly, we have improved the analysis of the trajectories of countries on the plane $X_{c,1} - X_{c,2}$ along the knee-like shape, clarifying some aspects of the transition across the three regimes, see SI, S1.2.

[l] Lucińska, M., & Wierzchoń, S. T. (2018). Clustering based on eigenvectors of the adjacency matrix. *International Journal of Applied Mathematics and Computer Science*, 28.

[m] Mealy, P., Farmer, J. D., & Teytelboym, A. (2018). A New Interpretation of the Economic Complexity Index (February 4, 2018). Available at SSRN: <https://ssrn.com/abstract=3075591> or <http://dx.doi.org/10.2139/ssrn.3075591>

9. Fig 1: in panel b I would label the outliers and comment. Also I would add a comment on why the residuals seem to be mostly skewed upwards. Does this help answering to point 2.3 of my comments? Along the same lines, I would also comment on the residuals of ECI vs $X_{c,2}$. These are clearly simply due to the choice of how to build the similarity matrix (being the method perfectly identical to ECI): can we gain some interpretation about the validity of this choice?

As the reviewer pointed out, these comments and the above ones are related one to the other. Resuming:

- In panel b the outliers are related to the interpretation of the matrix \mathbf{N}^A as a proximity matrix, which implies setting to zero the diagonal values. More explanations on this issue are reported in Response reported above (page 4 – 5, Figure 2);
- In panel c the presence of the scatter is due to the differences in the matrices from which the second eigenvector is computed. There is perfect coincidence between the term $X_{c,2}^A / \sqrt{k_c}$ and ECI, with the second eigenvector being computed from the matrix \mathbf{N}^A ; while there is good (but not perfect)

correlation between $X_{c,2}^B/\sqrt{k_c}$ and the vector ECI, as shown in panel c, when the eigenvector is computed from the matrix \mathbf{N}^B . As explained in the text (line 86) we find no definitive arguments to prefer using \mathbf{N}^A or \mathbf{N}^B as the base for our analysis, except for the fact that with \mathbf{N}^A the information carried by the first eigenvector is the same as diversification (in fact, $X_{c,1}^A$ equals $\sqrt{k_c}$).

Also, we thank the reviewer for the useful suggestion of adding some comments about the outliers, comments that we included at line 138 of the revised manuscript as defined above in this reply letter, page 5.

10. lines 225-227: “As such... countries.” I don’t agree with this statement. All the economic complexity metrics are essentially more or less sophisticated ways of measuring topological features of the locations-activities networks (see minor comment 1 below). While mathematically these metrics can have better or worse properties, it is only by comparing them against other sources of information that one is able to understand the actual value of the metrics. If we know these metrics do we know something more about the state of the system? Are we able to make better predictions? Or to do them with less assumptions? Less or noisier data?
11. Following my previous comment: can the authors give some ideas of how does GENEPLY compare to some standard macroeconomic features of countries (e.g. GDP)? Is there a reason to believe that GENEPLY is actually carrying more information about these than Fitness or ECI?

We thank the reviewer for these useful insights about our work. In this work our aim is to show that the two most used measures of EC, Fitness and ECI, can be reconciled in a unique measure of complexity, the GENEPLY. With the sentence the reviewer is referring to we highlight that the GENEPLY embeds two variables, i.e., the two eigenvectors, that can be used to trace the trajectories of growth of countries as driven by economic complexity in a 2D plane, without invoking other macroeconomic variables. The idea lying at the foundation of economic complexity is to find quantitative metrics that can complement the more standard ones in describing wealth and economic growth. In this sense, the comparison of these metrics to more standard ones (e.g., per capita GDP) can be a useful exercise, but it leaves much room to interpretation and discussion. Should one be happy of finding a high correlation of econocomplexity metrics with GDP pc (because this entails robustness of the methods, see [a,b,i]), or in contrast a low correlation coefficient, should be seen as good news (because this would be taken as a clue of having added independent information to the system)? Some of the angry debate which have characterized the economic complexity field in recent years have been based on these arguments: aiming at reconciling the contrasting views that emerged during these discussions, we deliberately decided to remain out of the arena.

We have edited the lines 243 – 246 as follows:

“...that most applications require. As such, the chance of maintaining the simplicity of a data driven approach endows the GENEPLY framework with the main founding-reason for which economic complexity was born, i.e., to provide the ground for a more quantitative, data-driven approach to the assessment of the economic growth potential of countries as guided by knowledge [n].”

Concerning with this argument, we added some comments in the Discussion section.

[n] Tacchella, A., Cristelli, M., Caldarelli, G., Gabrielli, A., & Pietronero, L. (2013). Economic complexity: conceptual grounding of a new metrics for global competitiveness. *Journal of Economic Dynamics and Control*, 37(8), 1683-1691.

Some minor comments:

1. In the abstract the authors state that Economic Complexity is about the diversity of product exports. While this was true for the original works, nowadays the field has extended much beyond, and the more general subject of EC can be better described as the diversity of (human) activities. Especially in the field of economics EC has been often confused as a framework to study global trade, and I think it is important to not further fuel this misunderstanding.

Good observation. The aim of the sentence the reviewer is referring to (“...economic complexity metrics, based on the diversity and sophistication of the products countries export”) was to sum up in few words the core of the metrics, but we agree that nowadays this sentence could be misleading and shallow. We edited the abstract in the following way:

“...economic complexity metrics, aiming at uncovering the productive knowledge of countries”.

2. line 179: does it really make sense to multiply an intensive complexity measure by the population of the country? What does that mean?

As the reviewer also pointed out in the previous comments, EC metrics (intensive complexity measures) are typically compared to standard (intensive) economic features as the GDP per capita [a,b]). For the sake of comparison, since we adopted the methodology in [o], which computes the evolving barycenter of the world weighted by the GDP (absolute value), we proceeded in defining the barycenter of the world according to the GENEPEY of countries by multiplying the metrics for the size of the countries’ population. We added a comment on this point at line 203:

“...for its population value in time, thus allowing for a fair comparison with the path followed by the GDP (in absolute value) in time.”

[o] Dobbs, R. et al. (2012). Urban world: Cities and the rise of the consuming class. Tech. Rep., McKinsey Global Institute .

To conclude, I have found this contribution very interesting at least in principle. However I think the manuscript in its present form does not allow to evaluate if this contribution adds substantially to the field, or if it just provides a neat mathematical framework but no additional forecasting or explanatory power. My comments are mostly in the direction of getting hints about this point (actually proving a better forecasting power would probably be beyond the scope of this paper). I don’t really expect that the authors perform all the exercises that I propose, I just intend them as a stimulus for the authors to find ways to better show the strength of their contribution.

We thank the reviewer for these comments and insights, which we think have been properly implemented. As for the forecasting and explanatory power, we already commented on the intrinsic difficulty to find an agreed choice of the variable(s) that need to be forecasted or explained; in many cases the GDP pc is selected as the target, but this is partially counterintuitive within a framework trying to innovate data-based economics from its very foundations. In our view, some of the angriness in the debate that developed about this field in the scientific literature can be ascribable to a lack of definition of a clear target for these metrics. In this context, our approach does not only add neatness to the mathematical framework, but, most importantly, it let economic insights naturally emerge from the mathematics.

References

[A] A network analysis of countries’ export flows: firm grounds for the building blocks of the economy
G Caldarelli, M Cristelli, A Gabrielli, L Pietronero, A Scala, A Tacchella. PloS one 7 (10), e47278

[B] Measuring the intangibles: A metrics for the economic complexity of countries and products M Cristelli, A Gabrielli, A Tacchella, G Caldarelli, L Pietronero. PloS one 8 (8), e70726

[C] Nestedness in complex networks: Observation, emergence, and implications, MS Mariani, ZM Ren, J Bascompte, CJ Tessone. Physics Reports 813, 1-90

[D] Ranking species in mutualistic networks. V Domínguez-García, MA Muñoz. Scientific Reports (2015) 10.1038/srep08182 5, 8182

Reviewer #2 (Remarks to the Author):

The paper provides an interesting and novel insight on a series of previously proposed metrics to assess in a synthetic way the capability stock of countries. It surely contributes to the previous literature in a constructive way providing a general framework to better understand the differences of existing metrics and how to, potentially, improve them, especially the Fitness.

The paper is well written, it is extremely neat and pleasant to read.

We are glad to read these positive comments and we acknowledge this reviewer.

I would have only two major remarks:

- i) figure 2 and related analysis in the text: differently from the first part, I find the analysis a bit weak and the three regimes sound a bit anecdotal. The authors should make this part more quantitative. As a simple (and minimum) step in this direction, they should somehow smooth and aggregate the dynamics in their plane to show the emerging flow as did in [1,2,3] for instance and try to show the three regimes. The analysis would be even more robust and complete, if the authors would try an analysis in the spirit of what did in [4] where with a simple Granger causality analysis it is shown that there are different regimes of interaction between two variables in the perspective of a dynamical system - the Granger test is just a suggestion, it can be any other techniques which is considered by the authors appropriate to quantify the effects in this plane. We should expect that in the left bottom corner, X_1 and X_2 to be essentially unrelated - causality speaking -, in the middle part, they should be related and perhaps one is leading the other - this would be an interesting question to address -, top right part, I do not know what to expect a priori.

We thank the reviewer for the useful suggestion, which allowed us to improve the manuscript. We agree about the necessity to reinforce the analysis of Figure 2 of the main text. To make the analysis more informative, we followed a path different from the one suggested by this reviewer. In fact, the two components $X_{c,1}$ and $X_{c,2}$ evolve simultaneously, being both variables determined by the way the matrix N , and thus the trade, changes year by year. In this framework, it seems meaningless to investigate if one component is driving the other, because both of them descend from the same information.

However, we welcome the suggestion of the reviewer to provide a more quantitative approach to our analysis. To this aim, we have analyzed the trend of the trajectories of all the countries for which there are continuous data in time (154 countries). Figure 1 of this reply shows the resulting dynamics. The arrows connect the point located at the center of mass of $X_{c,1}$ and $X_{c,2}$ during the first 3 years of analysis (1995 – 1998) to the center of mass during the last 3 years (2014 – 2017).

Figure 1: The path followed by the countries in time. The arrows, identifying the countries displacement in time, show the path from the point located at the center of mass of $X_{c,1}$ and $X_{c,2}$ during the first 3 years of analysis (1995 – 1998) to the point corresponding to the center of mass during the last 3 years (2014 – 2017).

The figure shows that countries experience different dynamics depending on the region in which they fall. The different behaviors are better visualized in Figure 2, where, following the suggestion of the reviewer, we have grouped countries into classes depending on their starting $X_{c,1}$ values. In order to make the overall dynamics clearer, we defined overlapping classes of countries using a moving window of 20 countries per each class: (i) we ordered the countries (and respective scores of the eigenvectors in time) for increasing starting $X_{c,1}$ values; (ii) by defining each class through a window of 20 countries, we computed the resultant vector of the displacements of the countries falling in that class; (iii) we applied the resultant vectors to the barycenter of the starting points of the single vectors that fall into the class.

In Figure 2 we show the aggregated dynamics of countries along the knee-shape. The colors sort the vectors for their length, as normalized for the longest vector recorded (light blue identifies the shorter ones, light purple the longer ones). The light blue vectors on the bottom left part of the knee identify the Impasse: the dynamics of the countries in this area are here tangled, as shown by the horizontal displacement of the vectors. Notwithstanding the presence of some uplifting movements of the classes around the minimum point in $X_{c,1} = 0.05$, the countries within this area are stacked in this dynamic of poor diversification and complexity within the cluster of lower growth. As soon as countries reinforce their knowledge, the countries experience higher values of $X_{c,2}$ until these values approach to zero: here it starts the Bounce, where countries boost their diversification and complexity, turning cluster membership by joining the more economically grown countries' club and thus increasing the similarity in the export basket with them. The jump is highlighted by longer vectors in violet and light purple. Once the economies have experienced the boost, they join the Arena of competition, for which continuous growth is determined. It is interesting to observe a divergent direction in the highest part of the Arena (high values of $X_{c,1}$ and $X_{c,2}$). Here, countries may lose ground on the plane of growth. There are many factors which may contribute to these downgrading dynamics. In fact, as described in the manuscript, the economic and financial crisis are more likely to be the cause of these drops; also, the entrance in the markets of new economies decreases the potential of economies to increase their economic complexity in time.

Figure 2: The time regimes of economic growth according to the two contributions $X_{c,1}$ and $X_{c,2}$. During time, countries move along the knee-shape designed by the arrows.

We added some comments on this part at line 163 and on and included this analysis in the SI, S1.2:

“... One recognizes that also the ensemble of the trajectories is knee-shaped: in fact, in each year of analysis the positions of countries in the plane $X_{c,1} - X_{c,2}$ configures in a knee-like shape as shown in Figure 1 for the year 2017. The presence of this shape is related to linear algebra and network science (see SI, S1.1). By analysing

the aggregated displacements of countries in time from to 1995 to 2017 (for details see SI, S1.2, Figure S8) it is possible to identify in the graph three regimes of growth:”

Line 168 “Impasse: the countries that lie within this area averagely exhibit a horizontal displacement dynamic, within the borders delimited by low values of ...”

Line 174 “Bounce: marked by the crossing of the zero value of the y-axis, this area defines the increment in quantity and quality of the exports. Here, the average dynamics of the countries is uplifting toward higher stages of growth. Countries ...”

Line 178 “Arena: ... competitiveness, where the GENEPY index of some countries increases in time, that of others follows a decreasing path, instead. In fact, in this area countries aim at increasing the ... higher scores in $X_{c,2}$. However, the entrance of new countries in the competitive market is likely to affect other countries growth. This area ...”

- ii) the authors often state that the high correlation of two metrics is somehow implicitly a proof of the fact that they carry similar information. For instance, it is used to justify that the non-linear specification of the fitness is likely not needed. While, generally speaking, I essentially agree with the content of the paper, I strongly disagree with the authors on this specific point. A priori, we do not know in which part of the variables is stored the informative content, two variables might be 99% correlated but the difference in their forecasting power can be paradoxically huge and all concentrated in the residual part. Let us consider two variables $Z1 = aX + (1-a)Y1$ and $Z2 = aX + (1-a)Y2$, and $Y1$ is orthogonal to $Y2$ and $Y1$ is signal while X and $Y2$ are noise. If 'a' is close to 1, the two variables will be highly correlated but only $Z2$ will be useful to forecast/explain something, the signal-to-noise ratio will be surely weak, but only $Z2$ will have signal despite the high correlation with $Z1$. This is for instance the case of ECI and Fitness, they are correlated, but once used in practice to forecast or explain economic dynamics, the discrepancies distinguish the two variables much more than what the correlation would suggest. I would suggest to either downgrade the argument, which is leveraged several times or, more involved, to show that the linearized fitness and the non-linear have a similar predictive/explanatory power (see for instance last suggestion). Provided the correlation argument only, the conclusion cannot be drawn and the statement should be only descriptive. As a side comment, the authors should also show that the results of the correlation does not depend on the specific estimator, if they use Spearman or Kendall's Tau correlation, do the results change?

We thank the reviewer for this good observation. The example he proposes above to demonstrate the weakness of our argument on correlation depicts a very peculiar situation, a reasonable one in theory, but not very likely to occur in the real world, where signal and noise are, typically, inextricably mixed together. However, we accept to follow this reviewer on his ground, and better contextualize the example: in order to have a correlation coefficient 0.99, the example specifies to $Z1 = 0.99X + 0.01Y1$ and $Z2 = 0.99X + 0.01Y2$. Since $Y1$ is signal and X is noise, the predictive power of $Z1$ on $Y1$ (the “signal” in the reviewer’s example) will be 0.01, when measured in terms of correlation. Of course, no one would be interested, in real-world applications, in a variable with 1% correlation coefficient with another one: in fact, the correlation coefficient, as computed from finite-size samples, is itself a random variable, i.e., it is known with uncertainty. The probability distribution of the correlation coefficient R between two uncorrelated random variables follows a transformed student-t distribution with $(n-2)$ degrees of freedom [a]. With a sample size around 200 (number of considered countries) a correlation coefficient >0.01 (in absolute value) between two uncorrelated random variables is obtained in 91% of the cases [a]. This testifies that correlation 0.01 is non informative with sample size 200. Suppose in contrast $Z1 = 0.8X + 0.2Y1$ and $Z2 = 0.8X + 0.2Y2$; this would lead to 0.2 correlation between $Z1$ and $Y1$, which

is a statistically significant positive correlation with sample size 200. However, in this case Z1 and Z2 have a 0.8 correlation, if Y1 and Y2 are orthogonal. To summarize: if the a coefficient in the reviewer example is very close to 1, it is not true that Z1 will be useful to explain/forecast something, because a (1-a) correlation with Y1 implies Z1 is not statistically distinguishable from random noise-in terms of predictive power on Y1 – in samples of size around 200 .

Another way to support the same reasoning is by considering that real data are of course affected by measurement and reporting errors: it is sufficient to perturb the values in the sample with a 1% multiplicative noise to have a drop to a 0.99 correlation between a measure and itself (for example, between the nonlinear Fitness values measured from the uncorrupted and corrupted dataset). Unfortunately, trade data carry errors which, due to the presence of corrections and assumptions to model the costs, are typically even larger than 1% [b]. Also, data sanitation [c] introduces corrections that we expect will have a larger than 1% drop in correlation.

Last but not least, the correlation coefficient between the linearized and non-linear version of the algorithm further increases to a stunning 0.999 if one avoids setting to zero the diagonal values of the N matrix (Figure 3 of this reply). In fact, the linearized algorithm in Eqs (13) is solved by the computation of the eigenvectors of the matrix N, Eqs (15), which provide the linear values of Fitness by mean of the mapping in Eqs (14). Instead, as we explain, the GENEPI arises from the eigenvectors of the matrix N, this one being interpreted as a proximity matrix, i.e., the diagonal values are set to zero. Also, in the case of interpreting the matrices N as proximity matrix provides a very good correlation of the non-linearly and linearly computed values, as we show here in Figure 4. Still, one should be aware that these correlation coefficients are the result of linearization plus self-loops elimination, not linearization alone.

As for the suggestion to also use other measures of correlation or dependency, we address the analysis requested by the reviewer by computing the dependence analysis among the non-linear and the linear FC algorithm with three correlation coefficients: Pearson, Spearman and Kendall. While the Pearson coefficient determines the correlation across values, the Spearman's and Kendall's ones are ranking-based. Figure 3 – 4 show the correlation coefficients, in time, between the non-linearly and linearly computed values of Fitness, whether or not the diagonal values are left as computed, respectively. As Figure 3 shows, the Pearson and Spearman coefficients are always in the range of 99%, while slightly lower values result for Kendall's tau.

On this issue, a comment has been added at lines 75 and on, of the revised manuscript:

“... Surprisingly enough, comparing the terms $X_{c,1}^B$ and F_c/k_c , or $X_{c,1}^B * k_c$ and F_c , for the Fitness values – analogously $Y_{p,1}^B$ and $Q_p k'_p$ (or $Y_{p,1}^B / k'_p$ and Q_p) for the Quality values – this linearization preserves >98% of the information (independently of the kind of indicator of correlation chosen, Figure S2), thus questioning ...”

We point out that a small oversight has been done in the caption of Fig S2 in the Supplementary Information. The correlation coefficient we plotted in that figure is of the Spearman's kind, not of the Pearson's one as wrongly typed in the document.

One last comment concerning with the performance of the linearized algorithm with respect to the non-linear one – in the field of EC – relates to the good performances that the linearized form has in packaging the trade matrix, thus using the (linear) values of Fitness and Quality to reorder the elements of the matrix, as we show here in Figure 5. In fact, as the figure shows, there are no significant differences between the non-linear and the linear algorithm for maximizing the data-packing of the trade matrix.

Figure 3: Pearson's (green), Spearman's (orange) and Kendall's (blue) correlation coefficients among the non-linearly and the linearly computed values of Fitness. The Spearman and Kendall's coefficients are ranking-based, while the Pearson's one compares the values between the two vectors. The results refer to the case in which the diagonal values of the matrix \mathbf{N} are left as computed.

Figure 4: Pearson's (green), Spearman's (orange) and Kendall's (blue) correlation coefficients among the non-linearly and the linearly computed values of Fitness. The Spearman and Kendall's coefficients are ranking-based, while the Pearson's one compares the values between the two vectors. The results refer to the case in which the diagonal values of the matrix \mathbf{N} are set to zero.

Figure 5: Nestedness capability – comparison among the performances of the non-linear FC algorithm (panel a) and its linearized form, (panel b). The matrix is the incidence matrix of the countries-products bipartite network during 2017.

As regards the results we present in this work, we are aware that the ground for comparison among the metrics of economic complexity is their performance in predicting some econometrics about the growth performance of countries (such as the GDP per capita).

In this work our aim is to show that the two most used measures of EC, Fitness and ECI, can be reconciled in a unique measure of complexity, the GENEPY. In this context, our approach does not only add neatness to the mathematical framework, but, most importantly, it naturally let economic insights emerge from the mathematics. The idea lying at the foundation of economic complexity is to find quantitative metrics that can complement the more standard ones in describing wealth and economic growth: in this sense, the validation of EC metrics against some economic indicator would induce a logical loop in the system, where the validation is based on the same variable subject to validation. This kind of comparison of EC metrics to more standard economic ones (e.g., GDP pc) can be a useful exercise, but leaving much room to interpretation and discussion. We therefore deliberately decided not to enter the validation arena in this work, trusting the validation efforts performed by others on the Fitness and ECI measures [d,e,f], and taking advantage of their results to also support our multidimensional metrics, whose components are in fact strictly related to Fitness and ECI.

[a] Kendall, M., & Stuart, A. (1977). *The advanced theory of statistics*. London: Griffin, 1977, 4th ed.

[b] Gaulier, G., & Zignago, S. (2010). *Baci: international trade database at the product-level (the 1994-2007 version)*.

[c] Morrison, G., Buldyrev, S. V., Imbruno, M., Arrieta, O. A. D., Rungi, A., Riccaboni, M., & Pammolli, F. (2017). *On economic complexity and the fitness of nations*. *Scientific reports*, 7(1), 1-11.

[d] Tacchella, A., Cristelli, M., Caldarelli, G., Gabrielli, A., & Pietronero, L. (2012). *A new metrics for countries' fitness and products' complexity*. *Scientific reports*, 2, 723.

[e] Hidalgo, C. A., & Hausmann, R. (2009). *The building blocks of economic complexity*. *Proceedings of the national academy of sciences*, 106(26), 10570-10575.

[f] Tacchella, A., Mazzilli, D. & Pietronero, L. (2018) *A dynamical systems approach to gross domestic product forecasting*. *Nature Phys* 14, 861–865.

A minor comment concerns figure 3, the one of the three different barycenters. The fact that the GDP barycenter is still far from the one provided by the GENEPI index is likely the sign that the potential of economic growth of Asian countries is still strong (even for China) as observed in [3].

This is a very useful comment, many thanks. We have included this observation in the manuscript at lines 207 and on:

"[...] poorly impacts the ability of countries to economically grow. The distance between the current position of the barycenter of GDP and GENEPI may also state that Asian countries (China included) still have a strong potential for economic growth, as also stated in [3]."

As a general suggestion, likely for the next paper - it would be interesting to test GENEPI index in a framework like the one proposed in [3] because this would be the true benchmark to see if GENEPI carries more information than Fitness and ECI and if the linearized part of the Fitness X1 is really carrying the same information of the non-linear counterpart.

We thank the reviewer for this suggestion. As explained above, the linear and nonlinear algorithms share the very same information, variations being statistically indistinguishable from noise in samples of size 200. Also, as for the predictive power of GENEPI, in this reply letter we have cleared that aiming at reconciling the contrasting views that emerged about EC metrics, we deliberately decided to remain out of the discussions about validation. However, we highlight that future work should be addressed toward the possibility of micro-founding the economic complexity theory thanks to the similarity of GENEPI and EXPI we have shown in this work.

[1] Pugliese et al. Economic Complexity as a Determinant of the Industrialization of Countries, 2017

[2] Cristelli et al., The heterogeneous dynamics of economic complexity, 2015

[3] Cristelli et al., On the predictability of growth, 2017

[4] Cristelli et al., The Virtuous Interplay of Infrastructure Development and the Complexity of Nations, 2018

REVIEWER COMMENTS

Reviewer #1 (Remarks to the Author):

I want to acknowledge the authors their extensive and convincing work in improving the manuscript.

In my opinion the manuscript in its present form is well written, clear and precise in discussing the advancements it proposes. I think that it does represent an important and substantial contribution to the field and therefore deserves publication in Nature Communications.

Andrea Tacchella

Reviewer #2 (Remarks to the Author):

I thank the authors for addressing the two comments. Concerning the first comment on robustifying the trajectories part, I think that now the paper has been improved significantly and the three regimes they find are more convincing.

Concerning the second, actually I disagree both with their reply and even more with the new sentences they added ' thus questioning the relevance of non-linearity to assess the Quality of goods and the Fitness of countries.'

The reason of my disagreement is based on the fact that I believe there is a small flaw in the reasoning proposed as a reply for the toy model I was suggesting. Let's go back to the model:

$$Z1 = a * X + (1-a) Y1$$

$$Z2 = a * X + (1-a) Y2$$

and the sake of simplicity X, Y1 and Y2 have unit variance and X is orthogonal to Y1 and Y2 and Y1 and Y2 are also orthogonal. Let's $b = 1-a$ for the sake of notation.

The first point I disagree on is that the correlation between Z1 and Z2 is not 'a' but rather $a/\sqrt{a^2 + b^2}$. Therefore a correlation of 0.98 is close to a scenario of $a = 0.90$ (the authors proved that for 200 countries $a = 0.8$ is significant and $a = 0.8$ implies a correlation of 97+%). I would therefore say that they are instead proving my point, we are in a regime of correlation where the 'a' is such that my example is statistically significant (or very close to be).

The second point is that they have 200 countries and 20 years. This means that the t-stat they estimated is the one for only one year, so the true significance over 20 years is the t-stat they estimate times $\sqrt{20}$ as a first approximation. This means that a scenario with 'b' much lower than 0.10 would be now significant, confirming again my point. That we can have high correlation, small 'a' and still discriminate between Z1 and Z2.

Third point, back to the toy model proposed, it is easy to realize that, assuming that Y1 is the component useful for the forecast, the best strategy is to build the variable Z1 - Z2, this one would magnify my signal to noise ratio in a dramatic if 'b' is small. This shows that they can be very correlated but the best scenario from a signal to noise ratio point of view stays in the difference of the variables not in the common part.

I really think that the paper is worth publishing but the authors have to remove the sentences they added and, as said, in my first review they have to mitigate the implicit statement that 'since the linear fitness and the non linear fitness are very correlated, then there are no evidences for going non linear.' High correlation does not imply same forecasting power and we are in regime where the toy model proposed shows that the component which is not common could be statistically significant as

discussed above. Actually the authors itself have shown that for $a = 0.8$ the results would significant having only 1 year of observation.

REVIEWER COMMENTS

Reviewer #1 (Remarks to the Author):

I want to acknowledge the authors their extensive and convincing work in improving the manuscript.

In my opinion the manuscript in its present form is well written, clear and precise in discussing the advancements it proposes. I think that it does represent an important and substantial contribution to the field and therefore deserves publication in Nature Communications.

We are grateful to read this positive comment and to have the approval for publication. We thank this Reviewer for the time dedicated to our work.

Reviewer #2 (Remarks to the Author):

I thank the authors for addressing the two comments. Concerning the first comment on robustifying the trajectories part, I think that now the paper has been improved significantly and the three regimes they find are more convincing.

Thanks a lot for your appreciation of our work, and for the kind suggestion provided in the first and in this review round: in fact, we agree that the revised version we submitted after the first review round is more convincing than the original manuscript, in particular in identifying the three regimes. Also, we are confident the additional variations we are now introducing, following this Reviewer's advice, are further improving the quality of the manuscript.

Concerning the second, actually I disagree both with their reply and even more with the new sentences they added ' thus questioning the relevance of non-linearity to assess the Quality of goods and the Fitness of countries.'

The debate which has emerged on the very significance of finding very high values of the correlation coefficient is very interesting in our view, but only partially relevant for the present manuscript. We are sorry for not having been able to separate, in our previous response, the statistical problem in itself (which is an intriguing and thought-provoking one) from its impact on our work (which, in contrast is limited, the whole issue being resolved by deleting or changing the tone of few sentences, as correctly stated by this Reviewer). After having received this comment, we therefore carefully scrutinized our manuscript to find out places where we might have left the reader with the impression of overselling our findings (i.e., that the linear and non-linear versions of the FC algorithm produce similar results, with correlation coefficient between the two being 0.995 on average). We have found four places where this might have occurred, at line 77, 140, 256 and 260. We have thus deleted from the manuscript the following sentences:

Line 77 "[...] thus questioning the relevance of non-linearity to assess the Quality of goods and the Fitness of countries."

Line 140 "This result confirms that Fitness values can be obtained from a linear equation to compute the Quality values (Figure S2)."

Line 256 "The linear nature of our framework on economic complexity, first hinted by the Authors in [12] and the success of its first eigenvector in reproducing the results of the non-linear FC algorithm (> 98%), seriously questions the necessity of a non-linear descriptor for economic complexity as stressed by the Authors in [15], instead."

and modified the sentence at line 260 as follows:

"The fact of having found very similar results between the linear and the non-linear versions of the FC algorithm cannot be systematically generalized to other cases: in fact, some bipartite systems may require a genuine non-linear approach to let their nested nature emerge. However, the good results obtained in this case suggest that there are also systems where non-linearity plays a minor role. We speculate that this might be related to the differences in the decision-making processes ruling these systems."

We believe the new version of the manuscript preserves all of its value after this modifications, with the additional advantage that on this specific issue we leave the reader with all the needed, quantitative, information (scatter plot in Figure 1 and values of the correlation coefficients), while we leave out the qualitative reasonings.

Our reply to this Reviewer could even stop here, since we have strictly followed the Reviewer's comment (more precisely, we considered this comment in a wider sense than requested, by extending to other sentences his demand) and "demined" our paper from possible misunderstanding deriving from this "correlation issue".

However, we believe scientific reviews, especially when written with a constructive purpose like in this case, also offer the ideal ground for discussing relevant scientific issues which in some cases, like the present one, even cross the borders of the specific paper being subject to review. With this intention, we add below some further considerations on the points raised by the referee, in an attempt to fully valorize the time and effort he has put in this review.

The reason of my disagreement is based on the fact that I believe there is a small flaw in the reasoning proposed as a reply for the toy model I was suggesting. Let's go back to the model:

$$Z1 = a * X + (1-a) Y1$$

$$Z2 = a * X + (1-a) Y2$$

and the sake of simplicity X, Y1 and Y2 have unit variance and X is orthogonal to Y1 and Y2 and Y1 and Y2 are also orthogonal. Let's b = 1-a for the sake of notation.

Thanks for summarizing the model. For the benefit of the Editor, we recall that in the model X and Y2 are random noise, while Y1 is signal. The scope of the model is to demonstrate that there might be cases when two variables (Z1 and Z2 here) are highly correlated (hence a is large), but the signal is embedded in one of the two variables (namely, Z1, through the effect of Y1) and not in the other.

The first point I disagree on is that the correlation between Z1 and Z2 is not 'a' but rather $a/\sqrt{a^2 + b^2}$.

Thanks for this comment, there was indeed some confusion in our previous reply. We try to remediate here with a more detailed description. For the implied variables we have:

$$E(X) = E(Y1) = E(Y2) = E(Z1) = E(Z2) = 0,$$

$$\sigma_X = \sigma_{Y1} = \sigma_{Y2} = 1,$$

where E() is the mean operator and σ is the standard deviation.

The correlation coefficient $\rho(g, h)$ between two generic variables g and h having null expected value is:

$$\rho(g, h) = \frac{E(g \cdot h)}{\sigma_g \cdot \sigma_h}.$$

Therefore, in this case, it holds:

$$E(X * Y1) = E(X * Y2) = E(Y1 * Y2) = 0$$

$$\sigma_{Z1} = \sigma_{Z2} = \sqrt{a^2 + (1 - a)^2}$$

$$E(Z1 * Z2) = a^2$$

$$E(Z1 * Y1) = 1 - a$$

$$\rho(Z1, Z2) = \frac{a^2}{a^2 + (1 - a)^2}$$

$$\rho(Z1, Y1) = \frac{1 - a}{\sqrt{a^2 + (1 - a)^2}}$$

Therefore a correlation of 0.98 is close to a scenario of $a = 0.90$ (the authors proved that for 200 countries $a = 0.8$ is significant and $a = 0.8$ implies a correlation of 97+%).

In our work we have found that the linear and non-linear versions of FC have a correlation coefficient which is 0.995 (as the average value over the available years). By setting $0.995 = \rho(Z1, Z2)$ we find $a = 0.935$. Under this scenario, the correlation between Z1 and Y1 is $\rho(Z1, Y1) = 0.069$.

The average correlation coefficient between the linear and non-linear version of the algorithms raises to 0.99985 (on average across the years) when the linear algorithm is implemented without setting the diagonal of the proximity matrix to zero (see Methods and Figure S2, Figure S5). If $\rho(Z1, Z2)$ is set to this value, in the toy model one has $a = 0.988$ and $\rho(Z1, Y1) = 0.012$.

I would therefore say that they are instead proving my point, we are in a regime of correlation where the 'a' is such that my example is statistically significant (or very close to be).

A p value 0.069 is not significant at the 5% level with respect to a null hypothesis of uncorrelation between two variables, in samples of 200 data. In fact, the critical value is 0.1166 [a, b]. The p -value corresponding to $\rho = 0.069$ is 0.17: in other words, there is a 17% probability of sampling a value larger than 0.069 in couples of samples of size 200 sampled from a bivariate distribution, with the two variables being uncorrelated [a, b].

Another way to visualize the same outcome is by numerically simulating the system of equations defining Z1 and Z2 (with $a = 0.935$) and plotting the resulting relation between Y1 and Z1. Figure 1 reports the results for 20 couples of samples of size 200. The resulting regression lines are reported in red.

Figure 1: In blue, the outcome of 20 numerical simulations of the system of equations defining Z1 and Z2 (sample size 200), obtained by setting $a = 0.935$, shown in terms of the variables Z1 and Y1. In red, the respective regression lines between Z1 and Y1 computed for each numerical simulation.

It is arguable from the graph that Z1 is quite uninformative for describing the 'signal', i.e. Y1, with several cases when the regression even assumes a negative slope (inverse relation with respect to the expected one).

This said, it is clear that each situation should be considered on its own, and the toy model proposed by the Reviewer is indeed a very useful tool for understanding the possible hidden relations among random variables. For examples, the correlation coefficients is 0.995 on average but drops to 0.98 in a single year: with $\rho = 0.98$, the correlation between Y1 and Z1 is indeed significant at a 5% level (p -value 0.045).

The second point is that they have 200 countries and 20 years. This means that the t-stat they estimated is the one for only one year, so the true significance over 20 years is the t-stat they estimate times $\sqrt{20}$ as a first approximation. This means that a scenario with 'b' much lower than 0.10 would be now significant, confirming again my point. That we can have high correlation, small 'a' and still discriminate between Z1 and Z2.

*Using t-stat values multiplied by $\sqrt{20}$ would imply the data in the 20 different years are independent. However, they are very far from being independent since a strong autocorrelation characterizes the Fc time series (both in their linear and in their non-linear). As a consequence, the equivalent size of the overall sample is not 200*20 as implicitly implied by the Reviewer, but much lower, each year carrying only a marginal addition of information with respect to the previous year. Still, we agree with this Reviewer that the effect of combining different years together should be studied in detail to draw definitive conclusions on the information carried by two variables with a large cross-correlation, especially with respect to their predictive power.*

Third point, back to the toy model proposed, it is easy to realize that, assuming that Y1 is the component useful for the forecast, the best strategy is to build the variable Z1 - Z2, this one would magnify my signal to noise ratio in a dramatic if 'b' is small. This shows that they can be very correlated but the best scenario from a signal to noise ratio point of view stays in the difference of the variables not in the common part.

Of course, this is true, but it looks like an ad-hoc strategy which applies by construction to the toy model. It could be interesting in the future to investigate if this strategy also turns out to be applicable to the real-world time situations.

I really think that the paper is worth publishing but the authors have to remove the sentences they added and, as said, in my first review they have to mitigate the implicit statement that 'since the linear fitness and the non linear fitness are very correlated, then there are no evidences for going non linear.' High correlation does not imply same forecasting power and we are in regime where the toy model proposed shows that the component which is not common could be statistically significant as discussed above. Actually the authors itself have shown that for $a = 0.8$ the results would significant having only 1 year of observation.

Since this point is not crucial to our paper, we have followed (and extended) the Reviewer suggestion to remove the mentioned sentences and we hope we have now clarified better any doubts left from our previous reply.

[a] Kendall, M., & Stuart, A. (1977). *The advanced theory of statistics. Volume 2: Inference and Relationship* London: Griffin, 1977, 4th ed.

[b] Howell, D. C. (2010) *Statistical methods for psychology (7th ed.)*. Wadsworth, Cengage Learning.

REVIEWERS' COMMENTS:

Reviewer #1 (Remarks to the Author):

Having read the debate on the points raised by the other referee, I think that the paper further improved with the toning down of some statements regarding the importance of non-linearity.

Since this point has emerged as an important one, I would suggest the authors to also include more explicitly the results of the tests they made on biological networks as a response to my first review. At the moment they just mention that the very strong correlation between the linear and non linear method might be lost on other bipartite networks, but since they have tested this explicitly I think it would be informative for the reader to see the results.

Other than this I think that the manuscript deserves publication.

Andrea Tacchella

Reviewer #2 (Remarks to the Author):

I really thank the authors for the extended answer and for updating the paper according to my previous review. As mentioned by the authors, this is beyond the scope of the paper but I really hope to continue the scientific discussion on the significance of two very correlated signals more directly.

REVIEWERS' COMMENTS:

Reviewer #1 (Remarks to the Author):

Having read the debate on the points raised by the other referee, I think that the paper further improved with the toning down of some statements regarding the importance of non-linearity.

Since this point has emerged as an important one, I would suggest the authors to also include more explicitly the results of the tests they made on biological networks as a response to my first review. At the moment they just mention that the very strong correlation between the linear and non linear method might be lost on other bipartite networks, but since they have tested this explicitly I think it would be informative for the reader to see the results.

Other than this I think that the manuscript deserves publication.

We thank this reviewer for the useful suggestion. We have included the results about the biological networks in the Supplementary Information, specifically in Supplementary Figure 11 and discussed in Supplementary Note 5. We have added a reference to this results in the main text at line 253 and on:

"... in fact, some bipartite systems may require a genuine non-linear approach to let their nested nature emerge (see, e.g., the results pertaining with the pollinators-plants bipartite network in Supplementary Figure 11, discussed in Supplementary Note 5)."

Our gratitude goes to the reviewer for the positive response and for having helped us increase the value of our work.

Reviewer #2 (Remarks to the Author):

I really thank the authors for the extended answer and for updating the paper according to my previous review. As mentioned by the authors, this is beyond the scope of the paper but I really hope to continue the scientific discussion on the significance of two very correlated signals more directly.

Many thanks to this reviewer for this constructive discussion, which we also hope to be continued in some other scientific contexts. We acknowledge the reviewer for having helped us add value to our work thanks to the useful comments and suggestions.